# Boreal forest soil chemistry drives soil organic carbon bioreactivity along a 314-year fire chronosequence

Benjamin Andrieux[1], David Paré[2], Julien Beguin[2], Pierre Grondin[3], Yves Bergeron[1]

[1]NSERC-UQAT-UQAM Industrial Chair in Sustainable Forest Management, Forest Research Institute, Université du Québec en Abitibi-Témiscamingue, Rouyn-Noranda, QC, J9X5E4, Canada
[2]Natural Resources Canada, Canadian Forest Service, Laurentian Forestry Centre, Québec, QC, G1V4C7, Canada
[3]Ministère des Forêts, de la Faune et des Parcs, Direction de la Recherche Forestière, Québec, G1P3W8, Canada

*Correspondence to*: Benjamin Andrieux (benjamin.andrieux@uqat.ca)

**Abstract.** Following wildfire, organic carbon (C) accumulates in boreal forest soils. The long-term patterns of accumulation as well as the mechanisms responsible for continuous soil C stabilization or sequestration are poorly known. We evaluated post-fire C stock changes in functional reservoirs (bioreactive and recalcitrant) using the proportion of C mineralized in $CO_2$ by microbes in a long-term lab incubation, as well as the proportion of C resistant to acid hydrolysis. We found that all soil C pools increased linearly with time since fire. The bioreactive and acid-insoluble soil C pools increased at a rate of 0.02 MgC.ha$^{-1}$.yr$^{-1}$ and 0.12 MgC.ha$^{-1}$.yr$^{-1}$, respectively, and their proportions relative to total soil C stock remained constant with time since fire (8% and 46%, respectively). We quantified direct and indirect causal relationships among variables and C bioreactivity to disentangle the relative contribution of climate, moss dominance, soil particle size distribution and soil chemical properties (pH, exchangeable manganese and aluminum, and metal oxides) to the variation structure of *in vitro* soil C bioreactivity. Our analyses showed that the chemical properties of Podzolic soils that characterise the study area were the best predictors of soil C bioreactivity. For the O layer, pH and exchangeable manganese were the most important (model-averaged estimator for both: 0.34) factors directly related to soil organic C bioreactivity, followed by time since fire (0.24), moss dominance (0.08) and climate and texture (0 for both). For the mineral soil, exchangeable aluminum was the most important factor (model-averaged estimator: -0.32), followed by metal oxide (-0.27), pH (-0.25), time since fire (0.05), climate and texture (~ 0 for both). Of the four climate factors examined in this study (i.e., mean annual temperature, growing degree-days above 5°C, mean annual precipitation and water balance) only those related to water availability, and not to temperature, had indirect effect (O layer) or a marginal indirect effect (mineral soil) on soil C bioreactivity. Given that predictions of the impact of climate change on soil C balance are strongly linked to the size and the bioreactivity of soil C pools, our study stresses the need to include the direct effects of soil chemistry and the indirect effects of climate and soil texture on soil organic matter decomposition in Earth System Models to forecast the response of boreal soils to global warming.

## 1 Introduction

Soil is the largest terrestrial carbon (C) reservoir (Scharlemann et al., 2014) and a major source of uncertainty in ecosystem C predictions (Shaw et al., 2014). Therefore, an advanced mechanistic understanding of soil C processes needs to be investigated and integrated into forecast models to reduce uncertainties in global C-cycle feedback projections and to better predict the effects of global change on soil C reservoir (Bradford et al., 2016; Schmidt et al., 2011). The maintenance of the vast soil C reservoir is partly under microbial control (Cotrufo et al., 2013) and could respond to variations in environmental conditions (Davidson and Janssens, 2006). Hence, the C-quality temperature hypothesis states that more "recalcitrant" soil organic matter should have higher temperature sensitivity (Craine et al., 2010; Fierer et al., 2005). According to this hypothesis, it is important to distinguish the recalcitrant portion of the soil organic matter from the active portion in order to predict the impact of a rise in temperature on soil heterotrophic respiration. Furthermore, the bioreactive C fraction of soil organic matter is cycled on

time scales relevant to global warming. Thus, quantifying the size of the bioreactive soil C reservoir and understanding the controlling factors of soil C bioreactivity ($C_{BioR}$) is key to inform models of C cycle in face of and to better anticipate global warming.

Wildfire is a major natural disturbance in boreal forests that drives the ecosystem C balance (Bond-Lamberty et al., 2007; Kurz et al., 2013) and is known to impact several soil properties, including organic matter quantity and quality (Certini, 2005; Knicker, 2007). Key soil properties, some evolving following fire (e.g., soil acidity) and some not (e.g., particle size distribution), interact with climate and vegetation composition in complex causal direct and indirect relationships to regulate post-fire soil C accumulation (Andrieux et al., 2018). A saturation of soil C accumulation, especially for its recalcitrant portion, is often observed in soils when the rates of organic matter input to the soil are increased (Stewart et al., 2007; Hassink, 1996). Saturation of recalcitrant C is believed to come from the finite capacity of stabilization mechanisms in soils, such as chemical protection from the decomposition of soil organic matter by clay surfaces (Six et al., 2002). At steady state and when the accumulation of physico-chemically stabilized soil C occurs, soil C can only accumulate as non-protected C (Castellano et al., 2015) that is more prone to decompose quickly. However, the long-term patterns of change in soil C quality and the accumulation pattern of recalcitrant and bioreactive C pools as well as the mechanisms responsible for continuous accumulation or stabilization of soil C reservoirs are poorly known and have not been explicitly integrated into soil biogeochemistry (Luo et al., 2016). Most models of soil C dynamics divide soil organic matter into several conceptual pools and simulate decomposition as a first-order decay process (Luo et al., 2016). As part of this study, we characterized the acid-insoluble and bioreactive soil organic C pools ($C_{slow}$ and $C_{fast}$, respectively, expressed as stocks) that accumulate following wildfire. The acid-insoluble soil C fraction is assumed to be "recalcitrant" or resistant to biological degradation (Paul et al., 2006; Xu et al., 1997). Hereafter, we define $C_{BioR}$ as the proportion of C mineralized in $CO_2$ by microbes at constant temperature and constant water content over a long period of time as a relative measure of soil C lability (Laganière et al., 2015; Xu et al., 1997). Assessing the sizes of resistant and bioreactive soil C fractions through a fire chronosequence would help modelers to enhance the current and future C balance of landscape prone to wildfires.

Besides its direct role in C cycling, climate has been shown to be an indirect predictor of soil C storage (quantity and accessibility for microbial decomposition) through its effects on geochemistry (Doetterl et al., 2015). In addition, vegetation types determine the quantity, quality and vertical distribution of soil litter inputs, and so lead to differential mechanisms of soil C protection and stabilization (Jobbagy and Jackson, 2000; Laganière et al., 2017), with the moss stratum being a major source of soil C inputs in boreal ecosystems (Preston et al., 2006). Although many of these processes have been investigated separately, we are not aware of any empirical study so far that has quantified all these processes simultaneously and assessed the relative contribution of climate, time since fire (TSF), vegetation attributes and soil physico-chemistry to soil $C_{BioR}$.

The objectives of this study are to fill these knowledge gaps by quantifying changes in boreal forest soil bioreactive C stock with TSF (from 2 to 314 years), and disentangling the direct and indirect relative contributions of climate, moss dominance, soil particle size distribution and soil chemical properties (pH, exchangeable manganese (Mn) and aluminum (Al), and metal oxides) to soil $C_{BioR}$ across the spruce-feather moss bioclimatic domain in eastern North America. Focusing on the complex interplay between climatic and non-climatic factors, and their direct or indirect influence on soil $C_{BioR}$, we addressed the following questions: i) Does soil $C_{BioR}$ reservoir change with soil organic C accumulation observed with TSF?; and ii) To what extent do direct and indirect relationships among TSF, climate, physico-chemical soil properties and bryophyte dominance influence soil $C_{BioR}$? We framed our study within the state factor model of ecosystems (Amundson and Jenny, 1997), which emphasizes soil physico-chemical properties understood to be important to the pedogenesis of Podzolic soils (Schaetzl and Anderson, 2005) that occur on the sampled sites. From there, our general hypotheses are that once site factors such as overstory composition, surficial deposits and soil drainage are accounted for, as they were in the present study: 1) soil $C_{BioR}$ increases as forest stands get older, leading to a buildup of soil bioreactive C stock under the cold conditions of the boreal forest; 2) alternatively, if the bioreactive soil C stock reaches a new equilibrium because of rapid turnover, the proportion of bioreactive

C stock should decline as total soil C stocks increases with TSF; and 3) soil $C_{BioR}$ is primarily controlled by TSF and moss dominance in the O layer, and by soil physico-chemistry in the mineral soil (see section 2.3 for detailed hypotheses).

## 2 Materials and methods

### 2.1 Site selection, sampling design and fieldwork

To account for the effects of TSF and climate on soil C pools, we established sample plots across both a chronosequence and a climosequence (Fig. 1; see Andrieux et al. (2018) for a description of the study area). Using numeric forest inventory maps compiled by the Ministère des Forêts, de la Faune et des Parcs du Québec (MFFPQ), we selected stands with as many similarities as possible in terms of canopy composition (black spruce [*Picea mariana*] stands), surficial deposits (thick till) and mesic drainage conditions. The soils that develop under these cool and humid climate with acidic litter inputs typically belong to the Podzolic order (Table 1). In boreal forests, podzolic soils often have thick organic surface horizon (O layer), an eluviated A horizon (Ae) from where leached materials accumulate in the illuviated B horizon (Fig. S1). Within mesic drainage conditions, soil texture was quite variable (Table 1). These stands were overlaid with fire maps produced by the MFFPQ and other published dendrochronological surveys (Belisle et al., 2011; Bouchard et al., 2008; Cyr et al., 2012; Frégeau et al., 2015; Le Goff et al., 2007; Le Goff et al., 2008; Portier et al., 2016) to establish the chronosequence. We assumed that the black spruce canopy composition did not change significantly with time and that the forest cyclically returned to a black spruce dominance after fire, in so-called recurrent dynamics, as previously described for these forests in a paleological survey (Frégeau et al., 2015). Then, while studying this ecosystem with a single and cyclic successional trajectory and low vegetation diversity, and by carefully selecting stable permanent site conditions, we guarded against pitfall conclusions associated with space–time substitutions when using a chronosequence approach (Walker et al., 2010; Kenkel et al., 1997).

For field inventory and soil sampling, we followed Canada's National Forest Inventory ground plot guidelines (NFI, 2016). Stand biophysical description and soil sampling took place in a single 314 m² circular plot (10 m radius) in each stand. Slope inclination and orientation were recorded from the centre of each plot with a clinometer and a compass, respectively. Every 2 m along two orthogonal transects oriented following the main cardinal directions, and for a total of 20 records per plot, the thickness of the O layer was measured on a sample taken with a soil auger, and the dominant moss types (*Sphagnum* spp. or feather mosses) were identified using 400 cm² microplots (Fig. S1). After litter and green living mosses were removed, the O layer was sampled at the edge of the plot in three 400 cm² microplots that were spaced 15 m from each other, from which we extracted volumetric mineral soil samples (top 15 cm) with a metallic cylinder (ø = 4.7 cm, height = 15 cm). One soil pit was dug at the plot edge and at the same location where we sampled one of the three O layer samples, down to the bottom of the podzolic B horizon or to the bedrock when possible, for soil description, and to collect the mineral soil from 15 to 35 cm under the forest floor–mineral soil boundary, as well as in the top 15 cm B horizon with a metallic cylinder (ø = 4.7 cm). The significant stone content at one site prevented us from sampling the mineral soil from 15 to 35 cm, thus, no analyses could be provided for this layer of soil. Samples were kept in the dark and brought to the lab within 15 days for each region. They were kept in the dark at 2°C until processing during the fall.

The fieldwork was conducted in 2015, from 15 June to 8 September. The sampling effort covered 72 sites in black spruce stands where fire had burned 2 to 314 years ago. Climate data were interpolated at the plot level using BioSim v10.3.2 (Régnière et al., 2013) together with 1981–2010 climate normal series (http://climat.meteo.gc.ca/) from surrounding weather stations, and considering local slope attributes measured in the field as correcting factors (Régnière, 1996). Soil characteristics are summarized below (Table 1).

## 2.2 Laboratory analyses

### 2.2.1 Soil preparation

First, we prepared a composite of soil materials obtained from every sampled microplot (N = 3), by plot and soil layer (O layer or top 15 cm of mineral soil), to create representative samples for each layer in each of the 72 sample plots. O layer samples were sieved through a 6-mm mesh before being oven–dried (60°C), whereas mineral soil samples (either top 15 cm of the mineral soil, mineral soil from 15 to 35 cm or top 15 cm of the B horizon) were dried by air and passed through a 2-mm sieve. Bulk density was determined after weighing the dried samples, assuming there were no coarse fragments in the O layer, and corrected for fragments > 2 mm for the mineral soil. Part of each sample was retained for soil incubation. We used the < 2 mm fraction to determine pH, exchangeable cation and texture (mineral soil only for the latter). Finely ground sub-samples (< 0.5 mm) were used for C concentration, pyrophosphate extractable Fe and Al (top 15 cm of the B horizon only) and acid hydrolysis analyses.

### 2.2.2 Soil physico-chemistry

C concentration of each sample was analyzed by dry combustion (Skjemstad and Baldock, 2007) using a Leco TruMac (Leco Corp, St. Joseph, MI, USA). Exchangeable cations were extracted using a Mehlich-3 solution and were analyzed by inductively coupled plasma atomic emission spectroscopy (Ziadi and Sen Tran, 2007), using an Optima 7300 DV (Perkin Elmer Inc., Waltham, MA, USA). Pyrophosphate extractable Fe and Al (i.e., organically complexed metals; Mpy, hereafter defined as metal oxides) were extracted with a 0.1N $Na_4P_2O_7$ solution before analysis with the Optima 7300 DV (Courchesne and Turmel, 2007). O layer and mineral soil pH were determined in a soil:water solution by weights of 1:10 and 1:2 (Hendershot and Lalande, 2007), respectively, using a pH meter (Orion 2 Star). Particle size distribution of the mineral soil was assessed using a standard hydrometer method (Kroetsch and Wang, 2007).

### 2.2.3 Incubation settings

Soil incubation followed the method described in Paré et al. (2011). We prepared a total of 215 microcosms (72 sites x 3 soil layers minus one sample in the 15–35 cm mineral soil). We used 9 g of oven-dried O layer and 50 g of air-dried mineral soil. Dried soil was used to ensure that the initial incubation moisture conditions were similar. Soil samples were placed in 100-mL bottom–perforated plastic containers. The containers were previously filled with glass wool (to avoid material losses during moisture adjustments) and pre-washed with HCl (0.1 M) followed by deionized water. The microcosms were saturated with deionized water, drained for 24 hours at 2°C, and weighed to determine their water-holding capacity. Over approximately 50 weeks of experiments, microcosms were placed under constant air temperature (26°C) and humidity (100%) in a growth chamber and, when necessary, deionized water was periodically added to adjust soil moisture to 85% of the water-holding capacity. Except during $CO_2$ production measurements, each microcosm was stored in a 500-mL Mason jar kept open to maintain aerobic conditions and to prevent $CO_2$ accumulation to toxic levels. A rubber septum was installed on the metal lid for gas sampling when measuring $CO_2$ production.

### 2.2.4 $CO_2$ production measurements

$CO_2$ produced by each microcosm was measured periodically (at days 20, 26, 48, 64, 108, 126, 154, 227, 264 and 340 for the O layer and at days 8, 14, 21, 29, 36, 43, 51, 57, 72, 79, 86, 101, 113, 140, 203, 238 and 358 for mineral soil layers; Fig. S2), using a LI-6400 portable photosynthesis system (LI-COR®, Lincoln, NE, USA) connected to an $N_2$ carrier gas (LI-COR® Application Note # 134). The flow rate of the carrier gas was set to 100 mL.min$^{-1}$ using the gas flow meter FMA1812A (Omega Engineering, INC., Norwalk, CO, USA). Initial $CO_2$ measurements were taken directly after hermetically sealing a jar with a

metal lid and final $CO_2$ measurements were taken after 4 h to 24 h, depending on the soil layer and the progress of the

experiment; these measurements were carried out using a 2.5-mL or 10-mL (for O layer or mineral soil, respectively) air

volume, extracted from the jar headspace with a syringe through the rubber septum. This gas sample was injected through the

carrier gas into the LI-6400 infrared analyzer. $CO_2$ concentration ($\mu mol.mol^{-1}$) was predicted using the linear regression of a

sample's measured $CO_2$ peak against calibration curves obtained from benchmark gas ($CO_2$ at 800 ppm and 3,000 ppm). The

first measurement (initial $CO_2$ concentration) accounted for the $CO_2$ concentration of the ambient air when closing the jars.

This value was subtracted from the final $CO_2$ concentration to account solely for the $CO_2$ produced by the microcosm.

All data were subsequently standardized to a 24-hour period to provide a daily respiration rate and to calculate cumulative C

mineralization (Fig. S2) (Paré et al., 2006). In short, we applied the gas law to convert $CO_2$ concentration ($\mu mol.mol^{-1}$) to a C

mass basis, using a constant pressure at 101.3 kPa and the specific head space volume of each sample (total volume of the jar

minus soil volume and container). Cumulative respiration was calculated according to the following Eq. (1):

$$M_t = M_{t-1} + \frac{(R_t + R_{t-1})}{2} \times (t - t_{-1}), \tag{1}$$

where $M_t$ (mg $CO_2$-C) is the cumulative mass of C mineralized at time $t$, $R_t$ (mg $CO_2$-C.d$^{-1}$) is the daily respiration rate at time

$t$, and $t$ is the Julian day (d). $M_t$ was divided by the initial C mass (g) of each sample to compute the specific respiration rate

(Rs; mg $CO_2$-C.g$^{-1}$C$_{org}$). Then, dividing Rs by 10 gave the percentage of initial mass of soil C lost through microbial respiration,

or $C_{BioR}$.

**2.2.5 Acid hydrolysis**

We used acid hydrolysis as an index of biologically recalcitrant soil C (Xu et al., 1997), which has been proposed as an

indicator of a slow-cycling soil C pool (Paul et al., 2006). Hydrolysis was carried out by refluxing 2 g of soil with 50 mL HCl

(6M) brought to the boiling point on a hot plate. We used a two-hour reaction time because the majority of soil organic matter

is hydrolyzed during the first two hours and longer reaction times do not significantly change C release (Silveira et al., 2008;Xu

et al., 1997). Acid-insoluble residues were separated from hydrolysates by filtering the solution on inert paper filters, rinsed

three times with 50 mL of deionized water to remove any chlorine residues, oven-dried at 60°C overnight, and weighed before

C concentration analysis by dry combustion (see section 2.3.2). Based on the total C concentration of the acid-insoluble

residues and mass loss of the samples during the treatments, the hydrolysability (Plante et al., 2006) of a sample was calculated

based on the following Eq. (2):

$$C_{AI} = \left( \frac{[C_{AI}] \times M_{AI}}{[C_i] \times M_i} \right) \times 100 \tag{2}$$

where $C_{AI}$ is the percentage of the acid-insoluble C (%), *[C$_{AI}$]* and *[C$_i$]* are the C concentration of the acid-insoluble residues

and of the initial soil (%), and $M_{AI}$ and $M_i$ are the mass of the acid-insoluble residues and of the initial soil sample (g),

respectively.

**2.3 Ecological *a priori* hypotheses**

According to our third general hypothesis and to address the complex interplay among climatic and non-climatic factors, we

first selected the following environmental variables documented in the literature as being important drivers of soil $C_{BioR}$ and

pedogenesis of Podzolic soils: climate (temperature and water supply), soil texture, TSF, dominance of the moss functional

type, soil pH, and concentration of metal oxides and of exchangeable elements (Mn and Al) (Wiesmeier et al., 2019; Schaetzl

and Anderson, 2005). As in other ecosystems (Fierer et al., 2003; Salomé et al., 2010), the boreal forest soil microbial

community as well as the chemical and physical environment change with soil depth (Clemmensen et al., 2013; Hynes and

Germida, 2013), suggesting different drivers of the decomposition process in O- and mineral soil layers. Hence, for each of

the O- and mineral soil layers, we built two separate sets of *a priori* ecological hypotheses expressed as direct acyclic graphs

(DAGs) representing different causal relationships among environmental variables and soil $C_{BioR}$ (Fig. 2). Therein, we tested

the validity of four competing a priori ecological hypotheses represented by a DAG. This hypothetico-deductive approach, in which each *a priori* hypothesis was supported by ecological knowledge, allows for testing an alternative causal explanation in a falsifiable form as regards the underlying mechanisms of soil $C_{BioR}$ in the two soil layers. For each soil layer, the first hypothesis assumed only direct relationships between environmental variables and soil $C_{BioR}$ (hypotheses O1 and MIN1 in Fig. 2), such that they mirrored the widespread assumptions used in soil C prediction models based on multiple regression or ANOVA analyses. In addition, framed within Jenny's factor model of soil formation (Jenny, 1994), these baseline hypotheses assumed independence among environmental variables. Alternatively, we formulated *a priori* competing hypotheses in which both direct and indirect effects among variables and soil $C_{BioR}$ were explicit (hypotheses O2 and MIN2 in Fig. 2). Justifications for each *a priori* ecological hypothesis are listed below.

**2.3.1 Baseline hypothesis for the O layer, O1**

This hypothesis assumes that soils that have developed under cooler conditions limiting microbial activity should have a greater $C_{BioR}$ (Laganière et al., 2015) once temperature constraints have been removed. Rainfall under good drainage conditions (such as in this study) should promote greater decomposition rates, and hence lower $C_{BioR}$. Because wildfire induces polymerization and polycondensation of organic compounds, resulting in residues that are more resistant to biological degradation (Certini, 2005; Gonzalez-Perez et al., 2004; Knicker, 2007), TSF is expected to have a direct and positive effect on $C_{BioR}$ which was anticipated to increase with TSF. Soil pH also had a direct effect on soil bioreactivity because it regulates the microbial community (Fierer and Jackson, 2006) and is a key determinant of the decomposition process (Prescott et al., 2000; Zhang et al., 2008). We expected decreasing $C_{BioR}$ with decreasing pH because acidic soil conditions limit the activity of soil decomposers. Compared to *Sphagnum* spp., feather mosses are more palatable to microbes (Fenton et al., 2010; Lang et al., 2009), so we expected lower $C_{BioR}$ with more Sphagnum. Also, Mn availability has been shown to be a good predictor of boreal soil C stocks (Stendahl et al., 2017); hence, Mn being a co-metabolic compound of lignin degradation, we assumed that Mn has a direct positive effect on $C_{BioR}$.

**2.3.2 Alternative hypothesis for the O layer, O2**

As in hypothesis O1, TSF, pH, moss functional type and Mn had direct effects on C bioreactivity. However, this hypothesis differed from O1 in that TSF and moss dominance also had indirect effects on $C_{BioR}$ through changes in pH conditions. We expected decreasing pH with increasing Sphagnum spp. dominance because some physiological characteristics of these organisms lead to environment acidification (Andrus, 1986). Also, in the short term, fire modifies pH through the liming effect (Gonzalez-Perez et al., 2004; Knicker, 2007). In the long term, soils acidify with TSF as a result of vegetation regrowth, which involves the exchange of protons against cations to maintain the physiological electro-neutrality of the vegetation (Driscoll and Likens, 1982). Contrary to hypothesis O1, which postulated that climate and soil texture had direct effects on $C_{BioR}$, hypothesis O2 assumed that these drivers had only indirect effects on $C_{BioR}$ through their influence on moss dominance. Indeed, we expected that Sphagnum spp. would dominate over feathermosses under wetter conditions induced by greater precipitation, fined-texture soils holding more water, or both, because of their greater dependence to high soil water content.

**2.3.3 Baseline hypothesis for the mineral soil, MIN1**

This hypothesis assumes that there are only direct effects of environmental variables on $C_{BioR}$ in the mineral soil. Climate and pH directly control the decomposition process. As the binding of organic matter with the mineral phase has been recognized as an important mechanism of C protection against decomposition (Doetterl et al., 2015; Kaiser et al., 2002; Porras et al., 2017), we assumed that there would be direct effects of soil texture and metal oxide contents on $C_{BioR}$. In the first years following fire, the slow incorporation of charred residues from upper soil layers into the mineral soil could decrease the organic

matter quality (Johnson and Curtis, 2001), resulting in a decrease of $C_{BioR}$ with increasing TSF. Mn availability could directly modulate $C_{BioR}$ (see O1), and exchangeable Al could impede microbial decomposition when in excess (Kunito et al., 2016).

**2.3.4 Alternative hypothesis for the mineral soil, MIN2**

As an alternative to hypothesis MIN1, this hypothesis assumes that only TSF, pH, metal oxides and Mn/Al have direct effects on $C_{BioR}$. Additionally, pH is assumed to decrease with TSF because of the imbalance in nutrient uptake caused by aggrading vegetation. Also, exchangeable cations are dependent on pH (Sanborn et al., 2011), and the decrease in pH favours the creation of organometallic complexes impeding microbial decomposition (Buurman and Jongmans, 2005; Porras et al., 2017). Contrary to hypothesis MIN1, which assumed direct effects of climate and soil texture on $C_{BioR}$, this hypothesis assumes that climate and soil texture have only indirect effects on $C_{BioR}$. The indirect effect of climate on $C_{BioR}$ is mediated through its effect on mineral weathering (Doetterl et al., 2015) and the quantity of metal oxides leached from the upper soil layers (Schaetzl and Anderson, 2005). Compared to coarse-textured soils, fine-textured soils have more reactive surface sites that can bind additional Mn and Al ions (Petersen et al., 1996).

**2.4 Calculations and data analyses**

**2.4.1 Index of moss dominance**

In order to account for the effects of moss functional traits on $C_{BioR}$ of the O layer, we differentiated between *Sphagnum* spp. and feather mosses, since they have different ecophysiological characteristics (Bisbee et al., 2001), e.g., feather mosses decompose faster than Sphagnum spp. (Fenton et al., 2010; Lang et al., 2009). Based on Nalder and Wein (1999), we calculated an index of moss dominance (IMD) using the following Eq. (3):

$$IMD = \frac{O_{sph}}{O_{sph} + O_{pl} + O_h + O_{pt}} \tag{3}$$

where $O$ is the sum of occurrence of a species in the 20 microplots (see section 2.1), *sph*: *Sphagnum* spp.; *pl*: *Pleurozium schreberi* (Brid.) Mitt.; *h*: *Hylocomium splendens* (Hedw.) Schimp.; *pt*: *Ptilium crista-castrensis* (Hedw.).

Feather mosses dominate the moss stratum when the IMD tends toward 0, whereas *Sphagnum* spp. dominates the moss stratum when the IMD tends toward 1. Some sites (n = 5) that recently had fires did not have any moss species regrowth at the time of the fieldwork. For these sites, we set the IMD to 0.

**2.4.2 Soil C quality and bioreactivity**

First, we wanted to estimate variation in the size of the bioreactive and recalcitrant soil C pools ($C_{fast}$ and $C_{slow}$, respectively, expressed as stocks) with TSF. For each soil layer (O layer, top 15 cm of mineral soil and 15 to 35 cm of mineral soil), we scaled up to plot scale the cumulative proportion of C mineralized at the end of the incubations and the proportion of acid-insoluble C using Eq. (4) and Eq. (5):

$$C_{fast} = \frac{C_{BioR}}{100} \times C \times D_B \times h \tag{4}$$

$$C_{slow} = \frac{C_{AI}}{100} \times C \times D_B \times h \tag{5}$$

where $C_{fast}$ and $C_{slow}$ are the bioreactive and recalcitrant soil C pools (Mg.ha$^{-1}$), $C_{BioR}$ is the percentage of initial mass of soil C lost through microbial respiration (%), $C$ is soil C content (%), $DB$ is the bulk density (g.cm$^{-3}$), $h$ is the soil depth (i.e., mean depth based on 20 measurements per plot for the O layer; cm) and $C_{AI}$ is the acid-insoluble C fraction (%). Hereafter, the total C stock ($C_{tot}$), $C_{fast}$ or $C_{slow}$ pool size represents, within each plot, the sum of each C pool across all soil layers.

Secondly, in order to express the qualitative (relative) changes in soil C in relation to environmental variables, we used the proportion of initial mass of soil C lost through microbial respiration as an index of C lability (see section 2.2.4). In the

statistical analyses, we considered the whole mineral soil (in the top 15 cm and in the 15- to 35- cm layer) as a single soil layer by calculating the weighted mean by depth for all mineral soil variables.

### 2.4.3 Statistical analyses

First, we evaluated post-fire C stock changes in functional reservoirs (bioreactive versus recalcitrant) using the linear regression of C stocks against TSF. Preliminary analyses with generalized additive models and piecewise regressions did not show any significant non-linear or segmented relationships. Secondly, we quantified direct and indirect causal relationships among variables and $C_{BioR}$ using confirmatory path analysis with directional separation tests (Shipley, 2000a), according to the set of alternative *a priori* hypotheses (Fig. 2). Path analysis was used together with Fisher's *C* test (Shipley, 2000b) as a simultaneous test of independence for a model basis set (i.e., all non-adjacent pairs of variables defined as claims of independence). This led us to quantify how our data supported each hypothetical DAG and to identify whether some hypotheses would be rejected based on a robust statistical test (Shipley, 2009). Fisher's *C* statistic was compared with a $\chi^2$ distribution with 2k degrees of freedom (where k is the number of claims of independence in a model basis set). We rejected a causal model at the significance level $\alpha = 0.05$ when $p < \alpha$. Prior to analyses, we standardized (reduced and centered) all variables to quantify their relative contribution to the variability of soil $C_{BioR}$.

The fit of each DAG for every soil layer (O and mineral soil) was compared using a model selection approach together with the second-order Akaike information criterion (AICc) in order to account for small sample sizes (Shipley, 2013). For model selection, we used the relative AICc difference with the "best" model or relative weight (Symonds and Moussalli, 2010). To avoid having latent variables in the models, and because we had no *a priori* knowledge about which specific climate, texture or exchangeable elements should be used for testing the validity of each hypothesis, we used the cross-product of four climatic variables (mean annual temperature: MAT, growing degree-day above 5°C: GDD5, mean annual precipitation: MAP, water balance: WB), three soil texture variables (sand %, silt % and clay %), and two exchangeable elements (Al and Mn, only for mineral soil). Therefore, we tested 12 and 24 model combinations for each hypothesis/DAG involving the O layer and the mineral soil, respectively. Given that each soil layer had two alternative causal hypotheses, we then compared 24- and 48-candidate DAG models using a model selection procedure for O layer and mineral soil, respectively. Model-averaged estimates were calculated by multiplying each estimate within each model by the corresponding Akaike weight and by summing the resulting values across all models; this allowed all models to influence model-averaged estimates. By doing so, we guarded against making arbitrary decisions about which model should be considered. We used the "ggm" package to compute Fisher's *C* statistic (Marchetti et al., 2015). All calculations and statistics were made using the R software version 3.4.3 (R Core Team, 2017).

## 3 Results

### 3.1 Post-fire soil C pool size

Total soil C stock ($C_{tot}$, i.e., the sum of O layer and mineral C stocks in the top 35 cm), the size of the recalcitrant C pool and the size of the bioreactive C pool ($C_{slow}$ and $C_{fast}$, respectively) all increased linearly with TSF (Fig. 3a). A minimum $C_{tot}$ value of 63 MgC.ha$^{-1}$ was observed for a 100-year-old stand, which is close to the $C_{tot}$ value of 66 MgC.ha$^{-1}$ of the youngest (2-year-old) stand. A maximum $C_{tot}$ value of 305 MgC.ha$^{-1}$ was observed for a 283-year-old stand. On average, $C_{slow}$ size was 6-fold bigger than $C_{fast}$ size (Table 2). $C_{fast}$ values ranged from 5 MgC.ha$^{-1}$ to 25 MgC.ha$^{-1}$ for a 100-year-old stand and for a 91-year-old stand, respectively. $C_{slow}$ values were 29 MgC.ha$^{-1}$ and 175 MgC.ha$^{-1}$ for a 2-year-old stand and for a 91-year-old stand, respectively.

Using these simple linear trends, $C_{tot}$ accumulated faster than $C_{slow}$ and $C_{fast}$ ($F_{3,209} = 257.6$, $p < 0.001$; Table 2 and Fig. 3b). Our data indicate that the overall soil C quality did not vary quantitatively with TSF ($R^2 < 0.01$, $p \geq 0.81$ for both C pools;

Table 2) because the proportion of $C_{slow}$ and $C_{fast}$ remained constant over the timespan of the fire chronosequence (Fig. 3b). These general trends are mostly influenced by the size of the $C_{slow}$ and $C_{fast}$ pools of the O layer, being 5 times and 2.4 times larger than the top 35 cm of the mineral soil one (i.e., sum of the two mineral soil layers), respectively (Table 2; Fig.S3). $C_{slow}$ and $C_{fast}$ decrease with soil layers from the surface soil horizon (O layer) to the deeper mineral soil, both in absolute size and proportion (Table 2; Fig. S3). Consistently with the whole data set, the proportion of $C_{slow}$ and $C_{fast}$ do not vary quantitatively

with TSF for all the soil layers analyzed separately (Table 2). The size of the $C_{slow}$ pool increases linearly with TSF in the O layer only ($R^2 = 0.09$, $p = 0.01$), not in mineral soil layers ($p > 0.07$ for both mineral soil layers). The size of the $C_{fast}$ pool increases linearly with TSF in the O layer ($R^2 = 0.12$, $p = 0.003$) and in the top 15 cm of the mineral soil ($R^2 = 0.05$, $p < 0.05$), not in the deepest mineral soil layer from 15 to 35 cm ($p > 0.21$).

**3.2 O layer path analysis and model selection**

The causal structure of the baseline hypothesis O1, which assumes that there are only direct effects of covariates on $C_{BioR}$ in the O layer, was rejected for all candidate models (Fisher's $C$ statistic $> 45$, $p < 0.05$; Table 3). Instead, the data better supported the causal structure of alternative hypothesis O2 for all models (Fisher's $C$ statistic $< 31$, $p > 0.25$; Table 3), which indicates that indirect effects among covariates and $C_{BioR}$ in the O layer need to be accounted for in order to properly assess the variation structure in the data. The model selection procedure revealed that the data were best explained by one leading model (hereafter,

"best" model; Fisher's $C$ statistic $= 22.3$, $p = 0.67$; Table 3); this model was associated with O2, with MAP as the climate variable and clay content as the texture variable. The Akaike weight for this model (68%) was about eight times greater than the weight of the second most supported model (8%). The model-averaging procedure revealed that exchangeable Mn and pH of the O layer were the two covariates that had the strongest direct and positive effects on $C_{BioR}$ of the O layer (both with an averaged path coefficient, pc $= 0.34$, $p < 0.01$; Fig. 4 and Table S1). TSF was the second most important relative driver with a

significant direct and positive effect on $C_{BioR}$ (pc $= 0.24$, $p < 0.05$; Fig. 4 and Table S1). Moss dominance had no significant direct effects on $C_{BioR}$ (pc $= 0.08$, $p > 0.41$; Fig. 4 and Table S1). In addition, both TSF and moss dominance had indirect effects on $C_{BioR}$ through their influences on pH (TSF→pH: pc $= -0.32$, $p < 0.01$; moss dominance→pH: pc $= 0.30$, $p < 0.01$; Fig. 4 and Table S1). In addition, the model contained an indirect effect of climate (MAP) on $C_{BioR}$ through its direct and negative effect on moss dominance (pc $= -0.25$, $p < 0.05$; Fig. 4 and Table S1). We detected no any effect of texture (clay

content) of the mineral soil on moss dominance ($-0.01 <$ pc $< 0.05$, $p > 0.43$).
   By allowing all of the models (O1 and O2) to influence coefficient estimates, the model-averaging procedure indicated that the most important variables exerting a direct control over $C_{BioR}$ of the O layer were as follows, by decreasing importance: pH and Mn, TSF, and moss dominance (Table S1). Moreover, we could not detect any direct effect of climatic and texture variables tested in this study on $C_{BioR}$ in the O layer.

**3.3 Mineral soil path analysis and model selection**

   The causal structure of the baseline hypothesis MIN1, which assumed that there were only direct effects of covariates on the $C_{BioR}$ of the mineral soil, was rejected (Fisher's $C$ statistic $> 52$, $p < 0.05$; Table 4). Instead, the data best supported the causal structure implied by the alternative hypothesis MIN2 indicating that, similar to the O layer, indirect effects among covariates need to be accounted for assessing in order to properly assess variation in the $C_{BioR}$ of the mineral soil. The model selection

procedure revealed that the data were best explained by one leading model (hereafter, "best" model; Fisher's $C$ statistic $= 27.76$, $p = 0.27$; Table 4). This model was associated with MIN2, with WB as the climate variable, clay content as the texture variable, and Al as the exchangeable element variable. The Akaike weight for this model (47%) was about three times greater than for the second most supported model (14%). The model-averaging procedure revealed that exchangeable Al had the strongest direct and negative effect on the $C_{BioR}$ of the mineral soil (pc $= -0.32$, $p < 0.001$; Fig. 5). Metal oxide content (pc $= -$

0.27, $p < 0.05$) and pH (pc $= -0.25$, $p < 0.05$) were the second most influential drivers with significant and negative direct

effects on the $C_{BioR}$ of the mineral soil. TSF had a small positive direct effect on the $C_{BioR}$ of the mineral soil, but this relationship was not significant (pc = 0.05, p > 0.36). In addition, pH induced two indirect effects on the $C_{BioR}$ of the mineral soil, i.e., through its negative and direct effects on Al and Mpy (pH→Al: pc = -0.24, p < 0.01; pH→Mpy: pc = -0.34, p < 0.01). Clay content had an indirect effect on the $C_{BioR}$ of the mineral soil, through its direct and positive effect on exchangeable Al (pc = 0.17 p < 0.05). Also, water balance had a weak indirect effect on the $C_{BioR}$ of the mineral soil through its direct effect on Mpy (pc = 0.11, p = 0.07).

By allowing all the models (MIN1 and MIN2) to influence estimates, the model-averaging procedure indicated that the most important variables tested in this study and exerting a direct control over the $C_{BioR}$ of the mineral soil were as follows, by decreasing importance: exchangeable Al, metal oxide contents, pH and TSF (Table S2). Moreover, we failed to detect any direct effect of climate or mineral soil texture on $C_{BioR}$.

## 4 Discussion

### 4.1 Post-fire soil C quality

Most of the studies on post-fire soil C have focused on immediate or short-term responses, and found that fire affects soil C quality by creating profound changes in the structure of soil organic matter compounds through thermal oxidation (Certini, 2005; Gonzalez-Perez et al., 2004). By using a long-term chronosequence of TSF ranging from two to 314 years, our study provides new insights into the understanding of the trajectory of changes in soil C quality following fire, over hundreds of years. Our estimates of the size of fast- and slow-cycling soil C pools and our results indicate that *i)* both pools accumulate with TSF, and *ii)* the proportion of each C pool remains constant with TSF relative to total soil C stock (Fig. 2 and Table 1). These results do not necessarily imply that fire has no effect on soil C functional pools, because our chronosequence has a low resolution for the first few years following fire, but rather suggest that such changes, if present, are not long-lasting. Our results also highlight that the accumulation process of the bioreactive soil C reservoir does not reach an equilibrium, at least not in the first three centuries following fire. Instead, environmental conditions limiting decomposition, such as cold temperatures under a thickening O layer developed with TSF, could have slowed down labile C degradation and allow its accumulation (Kane et al., 2005). Our results also emphasize that changes in the size of the soil functional reservoirs with TSF are stratified within the soil profile. This pattern is consistent with the fact that fire impact on soil C stock is limited to surface soil horizons (Andrieux et al., 2018).

### 4.2 Control mechanisms of the soil carbon bioreactivity

This study shows that soil $C_{BioR}$ is driven by several climatic and non-climatic variables, some being common both for O layer and mineral soil, and others not, suggesting that different mechanisms may be involved in the control of the decomposition process in the O layer and in the mineral soil (Shaw et al., 2015; Ziegler et al., 2017).

#### 4.2.1 Soil carbon bioreactivity in the O layer

Our results suggested that pH and exchangeable Mn are important drivers of $C_{BioR}$ in the O layer. Boreal evergreen coniferous species generate high-lignin litter and forest floor layers (Laganière et al., 2017). This is reflected in the high proportion of acid-insoluble C of O layer samples (among all the O layer samples, mean ± sd = 73 ± 5%, Fig. S3). Therefore, soil C cycling in boreal forests depends on the capacity of microbes to depolymerize lignin. Microorganisms in the acidic soils of this ecosystem are dominated by fungi that use metalloenzymes–such as Mn peroxidases–to metabolize lignin (Pollegioni et al., 2015), or are white-rot fungi (Basidiomycota) equipped with enzymes that oxidize lignin (Cragg et al., 2015). Soil C stocks in the boreal forest humus layer have been found to be negatively correlated with exchangeable Mn availability (Stendahl et al., 2017). In our study, exchangeable Mn of the O layer was positively correlated with $C_{BioR}$, suggesting that increasing Mn

availability stimulates organic matter breakdown and that an Mn bottleneck in soil C cycling may be present (Kranabetter, 2019). We also observed direct and positive causal relationships between pH and $C_{BioR}$ of the O layer, indicating that acidic soil conditions limit soil C mineralization (Prescott et al., 2000). Bacterial respiration and microbial community composition were found to be strongly determined by soil pH in the forest soil (Bååth and Anderson, 2003). We found that pH of the O layer decreased with TSF. Alongside the direct and positive effect of TSF on $C_{BioR}$ of the O layer, our results indicate that dynamic processes constrained by chemical soil properties shifting with stand development after burning (e.g. pH) drive the nature of soil organic matter and potentially the rate of C losses by heterotrophic respiration from boreal forest soils.

Altogether, these results emphasize the need to include both soil chemistry and biological mechanisms into models of soil C cycling to better anticipate the role played by boreal forest in C cycle-climate feedbacks. Soil C cycling in mechanistic models of forest C dynamics often assumes that climate drives decay and the transfer rate of and between soil C pools (see Deluca and Boisvenue (2012)). Based on our results, we argue that chemical drivers of soil organic matter decomposition, such as exchangeable Mn concentrations and pH, might also be used to modulate soil C dynamics in such models, and we especially advise to accounting for temporal shifts in soil pH occurring with stand development.

We did not detect any direct effect of climate on soil $C_{BioR}$ in the O layer. This finding is consistent with the results of unchanged soil C stocks with *in situ* experimental warming worldwide (van Gestel et al., 2018). Furthermore, when synthesizing data of *in situ* experimental warming, Carey et al. (2016) found no change in soil respiration rate for warmed compared to control plots at the global scale, whereas changes were found to be significant for the boreal biome. The cumulative C mineralization of incubated soils in our study was not modulated by *in situ* temperature, which supports the results of Carey et al. (2016) for their entire dataset, but not for the boreal biome-restricted dataset. However, Carey et al. (2016) did not study soils from the Canadian Boreal Shield.

### 4.2.2 Soil carbon bioreactivity in the mineral soil

As in the O layer, our results highlight the role of pH as a regulator of $C_{BioR}$ in the mineral soil. In addition to having a direct effect on $C_{BioR}$, pH also had two indirect effects. The first indirect effect is through the stimulation of metal oxide production with increasing acidic conditions. We observed that low-pH conditions correlated positively with higher metal oxide contents, which in turn correlated negatively with $C_{BioR}$ in the mineral soil. This result is consistent with previous findings showing the role played by pH in mineral weathering and the preservation of C from decomposition through organo-metal complexation (Andrieux et al., 2018). The second indirect effect of pH on $C_{BioR}$ in the mineral soil is mediated through exchangeable Al only, not through Mn (Table S2). Microbes are vertically stratified within the soil column (Clemmensen et al., 2013; Ekschmitt et al., 2008; Hynes and Germida, 2013), with fungi populating the upper soil layers because of their greater need for metabolic oxygen compared with bacteria, which can more easily dwell in the less-oxygenated deeper soil layers. Our results suggest that, contrary to the O layer, oxidative depolymerization of lignin compounds mediated by Mn peroxidases may not be a major process for C cycling in the mineral soil (see above). Instead, the negative effect of pH on exchangeable Al, together with the negative effect of exchangeable Al on $C_{BioR}$ in the mineral soil, indicates that low pH conditions favor a greater exchangeable Al abundance, which in turn impedes organic matter decomposition. These findings are consistent with the observed pH-dependent Al toxicity that slowed microbial catabolic activities in acidic forest soils in Japan (Kunito et al., 2016) and in laboratory experiments (Wood, 1995). Our study goes one step further in that we show that exchangeable Al content is directly related to soil texture (especially clay content) in these podzolic soils. This supports the hypothesis that exchangeable Al bound to fine mineral particles, such as clay, might act as a source of stored Al that can be mobilized and complexed with C to impede decomposition.

Contrary to the O layer, we found that TSF was only weakly correlated to mineral soil $C_{BioR}$ and pH, and these relationships were not significant. We also found that the indirect effect of climate (correlation between water balance and metal oxides) on $C_{BioR}$ was marginal. These results indicate that effects of TSF (direct and indirect) and water availability (indirect) on $C_{BioR}$ are

restricted to surface organic horizons. Our results support the idea that properties of the organic layer are more likely to be affected by fire because they are directly exposed to surface heating (De Bano, 1990), and that the thick humus layer of boreal forest soils (Table 1) protects deeper soil layers from shifts in environmental conditions. The fact that black spruce roots mostly develop in the top soil (Yuan and Chen, 2010) could explain why we did not observe shifts in pH of the mineral soil with TSF.

### 4.3 Implication for carbon cycling and research needs

Theoretically, soil C dynamics can be predicted through a knowledge of soil C pool sizes, changes in inputs, and sensitivity to environmental factors (Luo et al., 2016). Understanding the bioreactivity of the large boreal forest soils C reservoir is key to predicting future global C cycle in the face of global warming. As illustrated in our study, some factors may act as C stabilization agents of soil C (i.e., metal oxides binding organic matter), while others may contribute to accelerating or slowing down the rate of soil organic matter biological processing (i.e., exchangeable Mn as a co-metabolic compound of lignin degradation; when in excess, toxicity of exchangeable Al for microbes; soil acidity regulating the microbial activity), and finally, others are related to the quality of organic matter inputs to the soil (i.e., type of moss flora; low-quality organic materials left after fire) or simply to the time require by the system to adjust and reach steady-state (i.e., causal relationships between TSF and pH, or pH and metal oxides). Our "best" models showed that the climatic conditions experienced *in situ*, expressed here as temperature and water availability, had no direct effect on *in vitro* soil $C_{BioR}$. Moreover, the indirect effects of climate on soil $C_{BioR}$ are limited to water supply factors, not to temperature. "Best" models also reveal direct and indirect effects on $C_{BioR}$ of both site properties and factors that evolve with TSF. Understanding and predicting changes in soil chemistry is therefore a key challenge that remains to be addressed in future works in order to improve our understanding of soil C balance with global change. Our results are in agreement with Davidson and Janssens (2006) and Davidson (2015), suggesting that improvements to ESMs may arise from integrating the long-term effects of climate on soil properties with the environmental constraints on microbiological degradation of soil organic matter.

The results of this study identified new pathways for the control mechanisms of soil $C_{BioR}$ that could help to predict the response of boreal forest soils to global change. While earth system models (ESMs) commonly focus on a temperature dependence of soil C decomposition (Bradford et al., 2016), our study showed, in agreement with Rasmussen et al. (2018), that key soil properties, because of their relationship to soil C bioreactivity, could improve ESMs for modeling soil C dynamics in relation to climate change. In particular, our study shows that predictive models need to include the direct effects of soil chemistry and the indirect effects of climate and soil texture on soil $C_{BioR}$. Moreover, while some factors (metal oxides, TSF) were found to affect both soil $C_{BioR}$ (this study) and soil C stocks (Andrieux et al., 2018), at the same time, other factors did not have such effects (types of mosses, pH). For example, moss dominance had a direct effect on C stock (Andrieux et al., 2018), but not on $C_{BioR}$ (this study) in the O layer.

The path analyses and model selection procedure used in our study have made it possible to distinguish direct from indirect effects of ecological drivers on soil C dynamics. We found that the local climate shaped soil $C_{BioR}$ indirectly through effects on moss dominance and on metal oxides, and that of the four climatic variables examined, only the variables related to water supply–and not temperature–significantly but indirectly affected soil $C_{BioR}$. This suggests that the forecasted increase of 11% precipitation by the end of this century in eastern North America (IPCC, 2013) would indirectly modulate soil C stocks (Andrieux et al., 2018) and soil $C_{BioR}$ (this study), together with the indirect effects of climate on the mechanisms of soil C stock and bioreactivity. How the boreal ecosystem C balance will evolve in the context of global change might be assessed through further research focusing on the changes in soil physico-chemical reactions pertaining to the mechanisms of soil organic matter decomposition and stabilization (Thornley and Canell, 2001).

## 5 Conclusion

Our study aimed to quantify the long-term post-fire changes occurring in the functional soil C pools, and to disentangle the direct and indirect relative contribution of climate, moss dominance, soil particle size distribution and soil chemistry (pH, exchangeable Mn and Al, and metal oxides) on soil $C_{BioR}$. Using a chronosequence approach, we show that labile and recalcitrant soil C pools both increased continually from 2 to 314 years after fire. Changes in the size of the bioreactive and of the acid-insoluble C pools with TSF were found to be stratified within the soil profile and reveal that fire impact on soil functional C reservoirs is limited to surface soil horizons. The main drivers of $C_{BioR}$ varied with the soil layer considered. The breakdown of organic matter in the O layer was constraint by pH and exchangeable Mn, while that of the mineral soil was dependent on exchangeable Al availability, metal oxide content and pH. Moreover, our results suggest that for both soil layers, the complex interplay among biogeochemical covariates needs to be accounted for to assess the variation structure of $C_{BioR}$, with climate (water supply parameters only, not temperature) having only indirect effects. We argue that the direct and indirect effects that covariates have on the bioreactivity of soil organic carbon need to be integrated in models simulating C dynamics. This is key to forecast the response of the enormous boreal forest soil carbon pool to global warming.

## Data availability

All data presented in this paper can be accessed online free of charge on Canada's Open Government website (https://open.canada.ca, DOI to be determined).

## Author contribution

BA, DP, YB and PG participated in writing the funding application and in designing the study. BA and PG did the fieldwork. BA and DP contributed to the lab work. BA and JB analyzed the data. BA prepared the manuscript with contributions from all co-authors.

## Competing of interest

The authors declare that they have no conflict of interest.

## Acknowledgements

This project was funded by Mitacs Acceleration grants IT05018 (FR11062 to FR11067). We would like to thank Catherine Bruyère, Cécile Remy, Arnaud Guillemard and Eric Beaulieu for field assistance. We are grateful to Danielle Charron and Pierre Clouâtre for helping with field logistics. We warmly thank Véronique Poirier, Jean Noël and Emeline Chaste for help with geomatics work, and Serge Rousseau for laboratory analyses as well as Cindy Shaw for insightful comments. We declare that we have no conflicts of interest.

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

Table 1: General characteristics of the sampling sites.

| Variable | | | Minimum | Maximum | Mean (± sd) |
|---|---|---|---|---|---|
| Mean annual temperature (°C) | | | -2.80 | 0.70 | -0.5 (± 0.8) |
| Growing degree-day above 5°C (°C) | | | 838 | 1290 | 1110 (± 110) |
| Mean annual precipitation (mm) | | | 861 | 1027 | 934 (± 52) |
| Water balance (mm) | | | 493 | 660 | 558 (± 46) |
| Particle size (%) | Sand | | 26 | 92 | 70 (± 11) |
| | Silt | | 5 | 63 | 24 (± 10) |
| | Clay | | 2 | 22 | 6 (± 3) |
| Soil thickness (cm) | | | 10 | 103 | 39 (± 15) |
| Soil group[a] | | | Podzol | | |
| pH | O layer | All age-class (year) | 3.3 | 4.2 | 3.7 (± 0.2) |
| | | [2;30[ | 3.4 | 4.1 | 3.9 (± 0.2) |
| | | [30;60[ | 3.5 | 3.8 | 3.6 (± 0.1) |
| | | [60;100[ | 3.3 | 4.2 | 3.7 (± 0.3) |
| | | [100;150[ | 3.3 | 3.9 | 3.6 (± 0.2) |
| | | [150;200[ | 3.5 | 3.9 | 3.7 (± 0.2) |
| | | > 199 | 3.3 | 3.8 | 3.5 (± 0.2) |
| | Mineral soil (top 15 cm) | All age-class (year) | 4.2 | 5.6 | 4.7 (± 0.2) |
| | | [2;30[ | 4.4 | 5.0 | 4.7 (± 0.2) |
| | | [30;60[ | 4.4 | 5.1 | 4.7 (± 0.2) |
| | | [60;100[ | 4.2 | 5.6 | 4.7 (± 0.3) |
| | | [100;150[ | 4.4 | 4.9 | 4.6 (± 0.2) |
| | | [150;200[ | 4.4 | 4.7 | 4.7 (± 0.3) |
| | | > 199 | 4.3 | 5.2 | 4.6 (± 0.2) |
| | Mineral soil (from 15 to 35 cm) | All age-class (year) | 4.5 | 5.9 | 5.2 (± 0.3) |
| | | [2;30[ | 4.8 | 5.5 | 5.2 (± 0.2) |
| | | [30;60[ | 4.7 | 5.9 | 5.1 (± 0.4) |
| | | [60;100[ | 4.6 | 5.8 | 5.3 (± 0.3) |
| | | [100;150[ | 4.5 | 5.7 | 5.2 (± 0.3) |
| | | [150;200[ | 4.9 | 5.5 | 5.2 (± 0.2) |
| | | > 199 | 4.8 | 5.6 | 5.1 (± 0.2) |
| Bulk density (g.cm⁻³) | O layer | All age-class (year) | 0.05 | 0.15 | 0.08 (± 0.02) |
| | | [2;30[ | 0.08 | 0.15 | 0.10 (± 0.02) |
| | | [30;60[ | 0.06 | 0.10 | 0.08 (± 0.01) |
| | | [60;100[ | 0.06 | 0.12 | 0.08 (± 0.02) |
| | | [100;150[ | 0.05 | 0.10 | 0.08 (± 0.01) |
| | | [150;200[ | 0.05 | 0.10 | 0.08 (± 0.02) |
| | | > 199 | 0.06 | 0.10 | 0.08 (± 0.01) |
| | Mineral soil (top 15 cm) | All age-class (year) | 0.73 | 1.60 | 1.06 (± 0.18) |
| | | [2;30[ | 0.85 | 1.44 | 1.14 (± 0.15) |
| | | [30;60[ | 0.83 | 1.32 | 1.11 (± 0.20) |
| | | [60;100[ | 0.73 | 1.60 | 1.02 (± 0.21) |
| | | [100;150[ | 0.78 | 1.21 | 0.99 (± 0.14) |
| | | [150;200[ | 0.83 | 1.30 | 1.07 (± 0.14) |

| | | | |
|---|---|---|---|
| | > 199 | 0.79 | 1.35 | 1.06 (± 0.16) |

| Layer | Age-class | | | |
|---|---|---|---|---|
| | > 199 | 0.79 | 1.35 | 1.06 (± 0.16) |
| Mineral soil (from 15 to 35 cm) | All age-class (year) | 0.59 | 1.45 | 1.02 (± 0.21) |
| | [2;30[ | 0.67 | 1.29 | 1.03 (± 0.18) |
| | [30;60[ | 0.83 | 1.45 | 1.10 (± 0.20) |
| | [60;100[ | 0.71 | 1.41 | 1.00 (± 0.23) |
| | [100;150[ | 0.59 | 1.33 | 0.98 (± 0.24) |
| | [150;200[ | 0.65 | 1.41 | 1.05 (± 0.21) |
| | > 199 | 0.62 | 1.32 | 0.98 (± 0.17) |
| O layer depth (cm) | All age-class (year) | 9.8 | 49.3 | 22.6 (± 8.4) |
| | [2;30[ | 9.8 | 36.1 | 17.2 (± 7.3) |
| | [30;60[ | 11.6 | 20.9 | 17.2 (± 3.2) |
| | [60;100[ | 13.7 | 49.3 | 24.8 (± 9.7) |
| | [100;150[ | 10.6 | 39.7 | 21.5 (± 8.3) |
| | [150;200[ | 18.3 | 33.7 | 26.3 (± 4.8) |
| | > 199 | 18.4 | 41.9 | 28.4 (± 6.8) |

[a]According to IUSS Working Group WRB (2015).

Table 2: Post-fire soil C pool size and accumulation rates.

| Pool | Unit | Layer | Mean (± sd) | Equation | $R^2$ | p-value |
|---|---|---|---|---|---|---|
| $C_{tot}$ | MgC.ha$^{-1}$ | All | 150.80 (± 49.91) | 120.32 + 0.280*TSF[a] | 0.20 | **< 0.001** |
| | | O layer | 80.37 (± 35.55) | 66.00 + 0.13*TSF | 0.09 | **0.011** |
| | | Mineral soil (0-15 cm) | 32.64 (± 14.72) | 26.82 + 0.05*TSF | 0.08 | **0.013** |
| | | Mineral soil (15-35 cm) | 39.05 (± 26.37) | 29.22 + 0.09*TSF | 0.08 | **0.020** |
| $C_{slow}$ | MgC.ha$^{-1}$ | All | 69.01 (± 25.79) | 56.16 + 0.118*TSF | 0.14 | **0.002** |
| | | O layer | 58.43 (± 26.82) | 47.65 + 0.099*TSF | 0.09 | **0.012** |
| | | Mineral soil (0-15 cm) | 7.42 (± 3.95) | 6.73 + 0.006*TSF | 0.02 | 0.280 |
| | | Mineral soil (15-35 cm) | 4.17 (± 3.61) | 3.13 + 0.009*TSF | 0.04 | 0.076 |
| | % of $C_{tot}$ | All | 46.40 (± 9.35) | 46.74 - 0.003*TSF | < 0.01 | 0.808 |
| | | O layer | 72.58 (± 5.21) | 71.97 + 0.006*TSF | <0.01 | 0.469 |
| | | Mineral soil (0-15 cm) | 23.29 (± 7.80) | 24.69 - 0.013*TSF | 0.02 | 0.268 |
| | | Mineral soil (15-35 cm) | 12.45 (± 9.19) | 12.86 - 0.004*TSF | <0.01 | 0.783 |
| $C_{fast}$ | MgC.ha$^{-1}$ | All | 11.47 (± 3.59) | 9.25 + 0.020*TSF | 0.21 | **< 0.001** |
| | | O layer | 8.26 (± 3.86) | 6.45 + 0.017*TSF | 0.12 | **0.003** |
| | | Mineral soil (0-15 cm) | 2.06 (± 0.62) | 1.86 + 0.002*TSF | 0.05 | **0.050** |
| | | Mineral soil (15-35 cm) | 1.35 (± 0.79) | 1.19 + 0.001*TSF | 0.02 | 0.214 |
| | % of $C_{tot}$ | All | 7.85 (± 1.67) | 7.87 – 0.0002*TSF | <0.01 | 0.935 |
| | | O layer | 10.55 (± 2.42) | 10.11 + 0.004*TSF | 0.02 | 0.258 |
| | | Mineral soil (0-15 cm) | 7.16 (± 2.83) | 7.42 - 0.002*TSF | <0.01 | 0.590 |
| | | Mineral soil (15-35 cm) | 4.19 (± 1.84) | 4.33 - 0.001*TSF | <0.01 | 0.646 |

[a]TSF: time since fire (yr$^{-1}$).

Table 3: Model fitness to the data for *a priori* hypotheses for the O layer. Models are sorted by increasing second-order Akaike information criterion (AICc).

| Hypothesis | Climate | Texture | C statistic | df | p | K | AIC$_c$ | ΔAIC$_c$ | W |
|---|---|---|---|---|---|---|---|---|---|
| | | | | | | | | | |

| Hypothesis | Climate | Texture | C statistic | df | p | K | AICc | ΔAICc | W |
|---|---|---|---|---|---|---|---|---|---|
| **O2** | **MAP** | **Clay %** | **22.30** | **26** | **0.67** | **11** | **48.77** | **0.00** | **0.68** |
| O2 | WB | Clay % | 26.50 | 26 | 0.44 | 11 | 52.97 | 4.20 | 0.08 |
| O2 | MAP | Sand % | 27.28 | 26 | 0.39 | 11 | 53.76 | 4.99 | 0.06 |
| O2 | MAP | Silt % | 28.55 | 26 | 0.33 | 11 | 55.03 | 6.26 | 0.03 |
| O2 | WB | Sand % | 28.93 | 26 | 0.31 | 11 | 55.41 | 6.64 | 0.02 |
| O2 | MAT | Silt % | 29.03 | 26 | 0.31 | 11 | 55.50 | 6.73 | 0.02 |
| O2 | GDD5 | Silt % | 29.04 | 26 | 0.31 | 11 | 55.51 | 6.74 | 0.02 |
| O2 | MAT | Clay % | 29.29 | 26 | 0.30 | 11 | 55.76 | 6.99 | 0.02 |
| O2 | MAT | Sand % | 29.43 | 26 | 0.29 | 11 | 55.91 | 7.14 | 0.02 |
| O2 | GDD5 | Sand % | 30.12 | 26 | 0.26 | 11 | 56.59 | 7.82 | 0.01 |
| O2 | WB | Silt % | 30.48 | 26 | 0.25 | 11 | 56.95 | 8.18 | 0.01 |
| O2 | GDD5 | Clay % | 30.56 | 26 | 0.25 | 11 | 57.03 | 8.26 | 0.01 |
| O1 | WB | Clay % | 46.66 | 30 | 0.03 | 7 | 62.44 | 13.67 | 0.00 |
| O1 | MAP | Clay % | 47.95 | 30 | 0.02 | 7 | 63.73 | 14.96 | 0.00 |
| O1 | WB | Sand % | 50.53 | 30 | 0.01 | 7 | 66.30 | 17.53 | 0.00 |
| O1 | WB | Silt % | 53.50 | 30 | 0.01 | 7 | 69.27 | 20.50 | 0.00 |
| O1 | MAP | Sand % | 54.23 | 30 | 0.00 | 7 | 70.01 | 21.24 | 0.00 |
| O1 | MAP | Silt % | 56.92 | 30 | 0.00 | 7 | 72.70 | 23.93 | 0.00 |
| O1 | GDD5 | Clay % | 57.79 | 30 | 0.00 | 7 | 73.57 | 24.80 | 0.00 |
| O1 | GDD5 | Sand % | 58.62 | 30 | 0.00 | 7 | 74.40 | 25.63 | 0.00 |
| O1 | GDD5 | Silt % | 58.89 | 30 | 0.00 | 7 | 74.67 | 25.90 | 0.00 |
| O1 | MAT | Clay % | 59.82 | 30 | 0.00 | 7 | 75.59 | 26.82 | 0.00 |
| O1 | MAT | Sand % | 61.19 | 30 | 0.00 | 7 | 76.97 | 28.20 | 0.00 |
| O1 | MAT | Silt % | 62.22 | 30 | 0.00 | 7 | 78.00 | 29.23 | 0.00 |

Note: *Hypothesis*: model name; *Climate*: climate variable; *Texture*: texture variable; *C statistic*: statistic for Fisher's *C* test; *df*: degree of freedom; *p*: p-value for Fisher's *C* test (when p < 0.05, the model is not supported by the data); *K*: number of free parameters; *AICc*: second order Akaike information criterion; Δ*AICc*: relative AICc difference with the model that best fitted the data (in bold); *W*: Akaike weight. *MAT*: mean annual temperature; *GDD5*: growing degree-day above 5°C; *MAP*: mean annual precipitation; *WB*: water balance.

Table 4: Model fitness to the data for *a priori* hypotheses for the mineral soil. Models are sorted by increasing second-order Akaike information criterion (AICc).

| Hypothesis | Climate | Texture | Exchangeable element | C statistic | df | p | K | AICc | ΔAICc | W |
|---|---|---|---|---|---|---|---|---|---|---|
| **MIN2** | **WB** | **Clay %** | **Al** | **27.76** | **24.00** | **0.27** | **14.00** | **63.26** | **0.00** | **0.47** |
| MIN2 | MAP | Clay % | Al | 30.18 | 24.00 | 0.18 | 14.00 | 65.68 | 2.42 | 0.14 |
| MIN2 | WB | Clay % | Mn | 31.15 | 24.00 | 0.15 | 14.00 | 66.65 | 3.39 | 0.09 |
| MIN2 | GDD5 | Clay % | Al | 31.78 | 24.00 | 0.13 | 14.00 | 67.28 | 4.02 | 0.06 |
| MIN2 | MAT | Clay % | Al | 32.21 | 24.00 | 0.12 | 14.00 | 67.71 | 4.45 | 0.05 |
| MIN2 | MAP | Clay % | Mn | 32.81 | 24.00 | 0.11 | 14.00 | 68.31 | 5.05 | 0.04 |

| | | | | | | | | | | |
|---|---|---|---|---|---|---|---|---|---|---|
| MIN1 | GDD5 | Sand % | Mn | | 52.77 | 30.00 | 0.01 | 7.00 | 68.55 | 5.29 | 0.03 |
| MIN2 | GDD5 | Clay % | Mn | | 34.34 | 24.00 | 0.08 | 14.00 | 69.84 | 6.58 | 0.02 |
| MIN2 | MAT | Clay % | Mn | | 34.57 | 24.00 | 0.08 | 14.00 | 70.07 | 6.81 | 0.02 |
| MIN1 | MAT | Sand % | Mn | | 54.44 | 30.00 | 0.00 | 7.00 | 70.22 | 6.96 | 0.01 |
| MIN1 | GDD5 | Clay % | Mn | | 54.78 | 30.00 | 0.00 | 7.00 | 70.55 | 7.29 | 0.01 |
| MIN1 | GDD5 | Silt % | Mn | | 55.22 | 30.00 | 0.00 | 7.00 | 70.99 | 7.73 | 0.01 |
| MIN2 | GDD5 | Sand % | Al | | 35.51 | 24.00 | 0.06 | 14.00 | 71.01 | 7.75 | 0.01 |
| MIN1 | MAT | Clay % | Mn | | 55.90 | 30.00 | 0.00 | 7.00 | 71.68 | 8.42 | 0.01 |
| MIN2 | GDD5 | Sand % | Mn | | 36.21 | 24.00 | 0.05 | 14.00 | 71.71 | 8.45 | 0.01 |
| MIN2 | WB | Sand % | Al | | 36.26 | 24.00 | 0.05 | 14.00 | 71.76 | 8.50 | 0.01 |
| MIN1 | MAP | Clay % | Mn | | 56.70 | 30.00 | 0.00 | 7.00 | 72.48 | 9.22 | 0.00 |
| MIN2 | WB | Sand % | Mn | | 37.14 | 24.00 | 0.04 | 14.00 | 72.64 | 9.38 | 0.00 |
| MIN1 | WB | Clay % | Mn | | 57.54 | 30.00 | 0.00 | 7.00 | 73.31 | 10.05 | 0.00 |
| MIN1 | MAT | Silt % | Mn | | 57.65 | 30.00 | 0.00 | 7.00 | 73.43 | 10.17 | 0.00 |
| MIN2 | MAT | Sand % | Al | | 38.06 | 24.00 | 0.03 | 14.00 | 73.56 | 10.30 | 0.00 |
| MIN2 | GDD5 | Silt % | Mn | | 38.19 | 24.00 | 0.03 | 14.00 | 73.69 | 10.43 | 0.00 |
| MIN2 | MAT | Sand % | Mn | | 38.44 | 24.00 | 0.03 | 14.00 | 73.93 | 10.67 | 0.00 |
| MIN1 | WB | Sand % | Mn | | 58.57 | 30.00 | 0.00 | 7.00 | 74.34 | 11.08 | 0.00 |
| MIN2 | GDD5 | Silt % | Al | | 39.04 | 24.00 | 0.03 | 14.00 | 74.53 | 11.27 | 0.00 |
| MIN1 | MAP | Sand % | Mn | | 60.15 | 30.00 | 0.00 | 7.00 | 75.92 | 12.66 | 0.00 |
| MIN2 | MAT | Silt % | Mn | | 41.32 | 24.00 | 0.02 | 14.00 | 76.82 | 13.56 | 0.00 |
| MIN2 | MAP | Sand % | Al | | 41.49 | 24.00 | 0.01 | 14.00 | 76.99 | 13.73 | 0.00 |
| MIN2 | MAP | Sand % | Mn | | 41.88 | 24.00 | 0.01 | 14.00 | 77.38 | 14.12 | 0.00 |
| MIN2 | MAT | Silt % | Al | | 42.64 | 24.00 | 0.01 | 14.00 | 78.14 | 14.88 | 0.00 |
| MIN2 | WB | Silt % | Mn | | 42.64 | 24.00 | 0.01 | 14.00 | 78.14 | 14.88 | 0.00 |
| MIN2 | WB | Silt % | Al | | 43.25 | 24.00 | 0.01 | 14.00 | 78.75 | 15.49 | 0.00 |
| MIN1 | WB | Silt % | Mn | | 63.72 | 30.00 | 0.00 | 7.00 | 79.49 | 16.23 | 0.00 |
| MIN1 | MAP | Silt % | Mn | | 65.01 | 30.00 | 0.00 | 7.00 | 80.79 | 17.53 | 0.00 |
| MIN2 | MAP | Silt % | Mn | | 47.03 | 24.00 | 0.00 | 14.00 | 82.53 | 19.27 | 0.00 |
| MIN1 | GDD5 | Sand % | Al | | 67.22 | 30.00 | 0.00 | 7.00 | 83.00 | 19.74 | 0.00 |
| MIN2 | MAP | Silt % | Al | | 47.77 | 24.00 | 0.00 | 14.00 | 83.27 | 20.01 | 0.00 |
| MIN1 | MAT | Sand % | Al | | 68.01 | 30.00 | 0.00 | 7.00 | 83.79 | 20.53 | 0.00 |
| MIN1 | WB | Sand % | Al | | 69.72 | 30.00 | 0.00 | 7.00 | 85.50 | 22.24 | 0.00 |
| MIN1 | MAP | Sand % | Al | | 72.48 | 30.00 | 0.00 | 7.00 | 88.26 | 25.00 | 0.00 |
| MIN1 | GDD5 | Silt % | Al | | 72.85 | 30.00 | 0.00 | 7.00 | 88.63 | 25.37 | 0.00 |
| MIN1 | MAT | Silt % | Al | | 74.41 | 30.00 | 0.00 | 7.00 | 90.19 | 26.93 | 0.00 |
| MIN1 | WB | Clay % | Al | | 74.57 | 30.00 | 0.00 | 7.00 | 90.35 | 27.09 | 0.00 |

| | | | | | | | | | | | |
|---|---|---|---|---|---|---|---|---|---|---|---|
| MIN1 | MAP | Clay % | Al | | 74.92 | 30.00 | 0.00 | 7.00 | 90.69 | 27.43 | 0.00 |
| MIN1 | GDD5 | Clay % | Al | | 75.10 | 30.00 | 0.00 | 7.00 | 90.88 | 27.62 | 0.00 |
| MIN1 | MAT | Clay % | Al | | 75.36 | 30.00 | 0.00 | 7.00 | 91.13 | 27.87 | 0.00 |
| MIN1 | WB | Silt % | Al | | 78.06 | 30.00 | 0.00 | 7.00 | 93.83 | 30.57 | 0.00 |
| MIN1 | MAP | Silt % | Al | | 80.53 | 30.00 | 0.00 | 7.00 | 96.31 | 33.05 | 0.00 |

Note: *Hypothesis*: model name; *Climate*: climate variable; *Texture*: texture variable; *Exchangeable element*: exchangeable element variable; *C statistic*: statistic for Fisher's *C* test; *df*: degree of freedom; *p*: p-value for Fisher's *C* test (when $p < 0.05$, the model is not supported by the data); *K*: number of free parameters; *AICc*: second order Akaike information criterion;
Δ*AICc*: relative AICc difference with the model that best fitted the data (in bold); *W*: Akaike weight. *MAT*: mean annual temperature; *GDD5*: growing degree-day above 5°C; *MAP*: mean annual precipitation; *WB*: water balance.

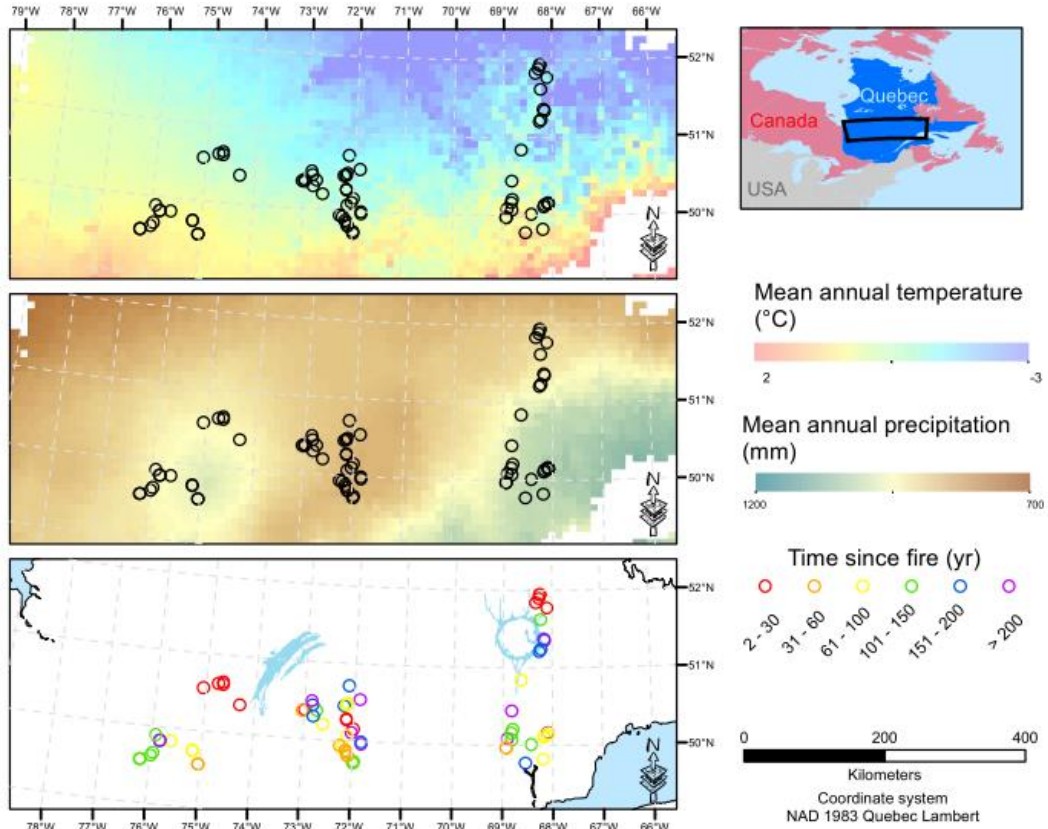

Figure 1: Map of the study area showing the location of the sample plots. Mean annual temperature (upper panel) and mean
annual precipitation (middle panel) are interpolations from the 1981–2010 Canadian climate normals on a 10 x 10 km pixel grid (Chaste et al. 2018 ). The lower panel presents the location of the sample plots in relation to time since fire (yr[-1]).

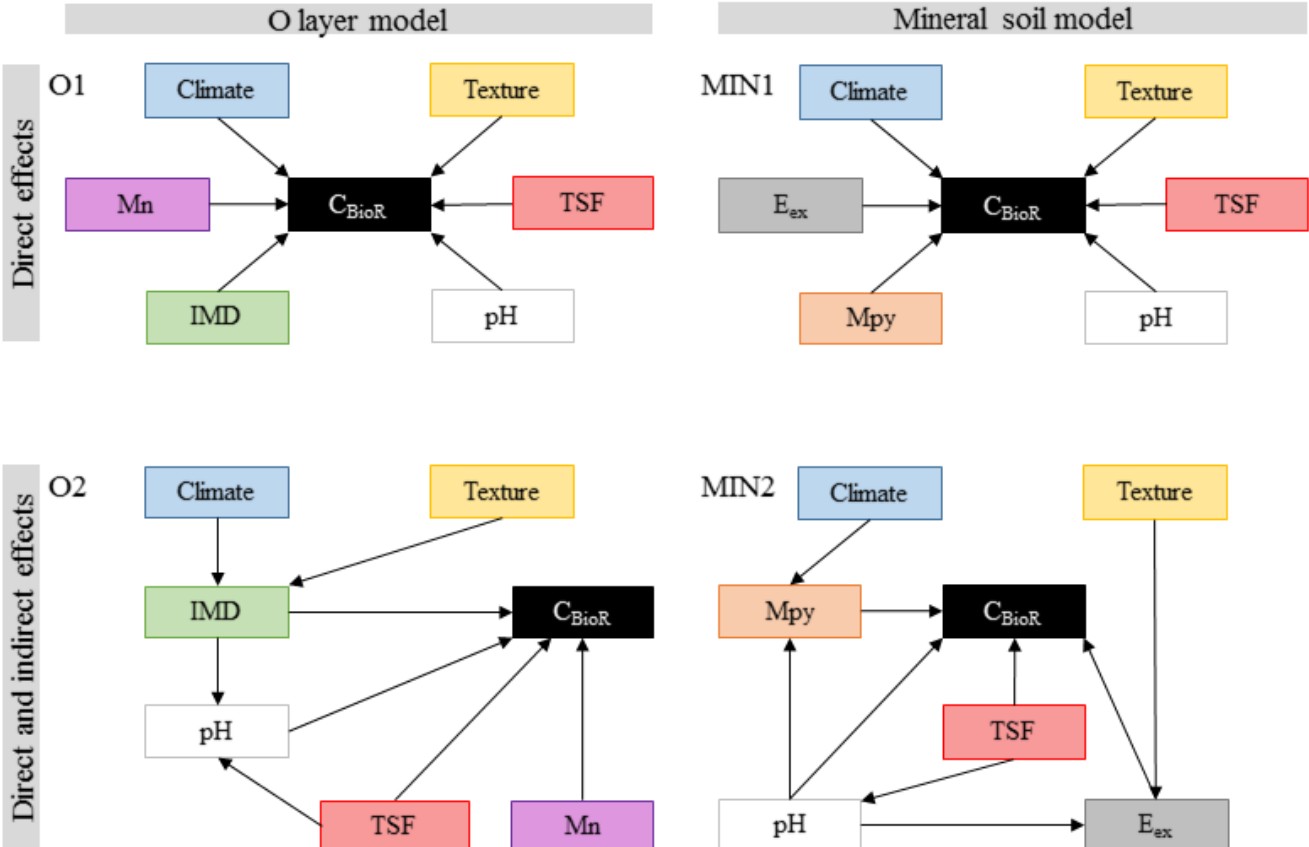

Figure 2: Path models for each of the multivariate causal hypotheses. Arrows indicate direct causal relationships. *Climate*: climate variable; *Texture*: texture variable; *TSF*: time since fire; *pH*: pH of the O layer (O1 and O2) or of the top 35 cm of mineral soil (MIN1 and MIN2); *IMD*: index of moss dominance; *Mn*: exchangeable manganese of the O layer; *Mpy*: pyrophosphate extractable metals; $E_{ex}$: exchangeable element (Al or Mn) of the mineral soil.


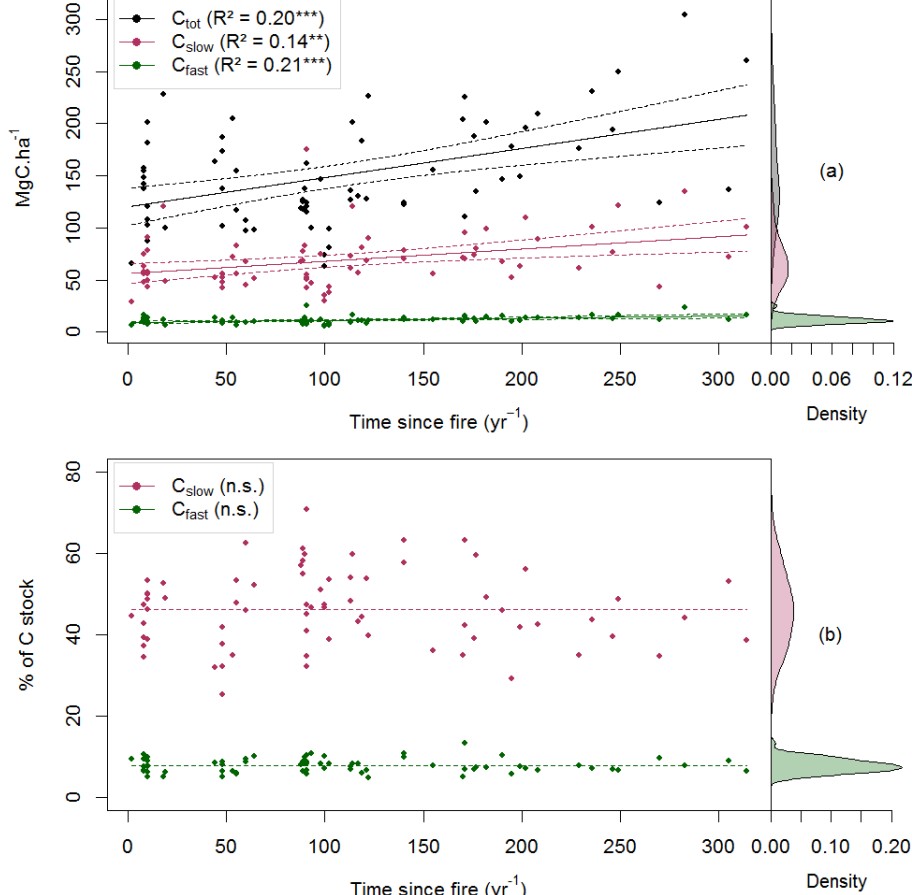

Figure 3: Carbon quality as a function of time since fire (TSF). The upper panel shows the total soil C reservoir ($C_{tot}$), the recalcitrant C pool ($C_{slow}$) and the bioreactive C pool ($C_{fast}$) sizes as a function of TSF (a, on the left), and the kernel density for each pool (a, on the right). The lower panel shows the proportion of $C_{slow}$ and $C_{fast}$ relative to $C_{tot}$ as a function of TSF (b, on the left), and their kernel density (b, on the right).

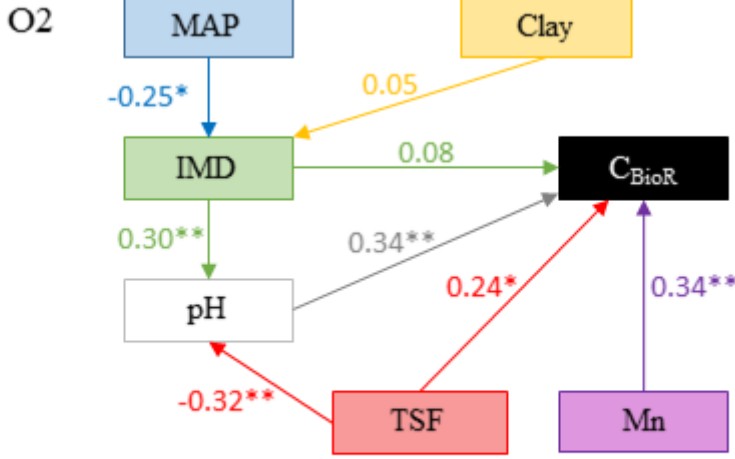

Figure 4: Model that best fitted the data to explain carbon bioreactivity ($C_{BioR}$) in the O layer. Arrows indicate direct causal relationships. The numbers are standardized averaged path coefficients obtained using model averaging (see Table S1 and text for further details). *MAP*: mean annual precipitation; *Clay*: clay content in the top 35 cm of mineral soil; *Mn*: exchangeable manganese; *pH*: pH of the O layer; *TSF*: time since fire; *IMD*: index of moss dominance. (*) $p < 0.05$; (**) $p < 0.01$.

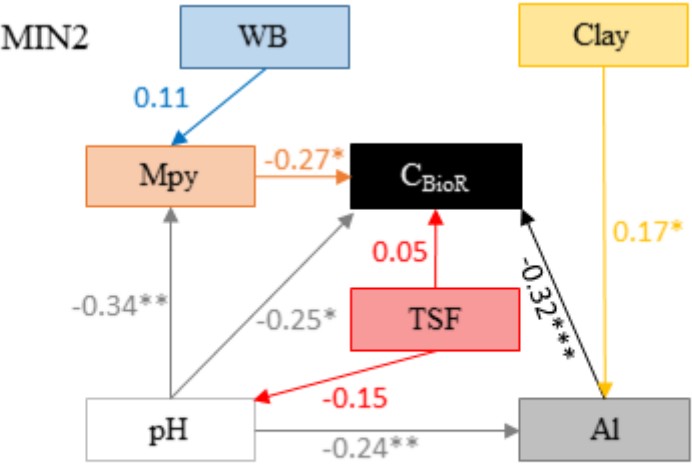

Figure 5: Model that best fitted the data to explain the carbon bioreactivity ($C_{BioR}$) in the top 35 cm of the mineral soil. Arrows indicate direct causal relationships. The numbers are standardized averaged path coefficients obtained using model averaging (see Table S2 and text for further details). *Al*: exchangeable aluminum in the top 35 cm of the mineral soil; *pH*: pH of the top 35 cm of the mineral soil; *TSF*: time since fire; *Mpy*: metal oxide content in the top 15 cm of the B horizon; *WB*: water balance; *Clay*: clay content of the top 35 cm of the mineral soil. (*) $p < 0.05$; (**) $p < 0.01$; (***) $p < 0.001$.
