# Peer review of "Boreal forest soil chemistry drives soil organic carbon bioreactivity along a 314-year fire chronosequence"

_SOIL, 2019_

## Referee Comment (RC1) · Anonymous Referee #1 · 20 Dec 2019

General comments In this study, the authors were evaluating post fire carbon stock changes in functional reservoirs (bioreactive and recalcitrant) using the proportion of C mineralized in $CO_2$ by microbes in a long-term lab incubation, as well as the proportion of C resistant to acid hydrolysis. Through the manuscript (already in Abstract) there are problems with abbreviations, one can find through the text carbon and C, bioradiactive C and CBioR, carbon dioxide and $CO_2$, etc. If you have started to use abbreviations, please be constant through entire text.

Introduction is informative, but I would expect more talk on the topic, why the C bioreactivity is important, and what does it mean if we have the changes in C bioreactivity

reservoirs through fire chronosequences. The hypothesis at the end of the introduction are OK, but when the other set of hypothesis are presented in Material and methods section, this creates some confusion. Study design needs some improvements (see my detailed comments), as currently it is difficult to understand how many microplots (for moss biomass measurements) were established per sample plot (maybe a scheme describing the measurements from sample plot would be useful as supplementary material). I also can't understand why the samples were incubated with so high temperature (+26°C), and only with one temperature. Usually, during the incubation, one attempts to mimic the field conditions (use temperatures similar to real soil temperature). And due to that I'm really concerned that are the cumulative respiration calculations actually valid. Results and Discussion section could benefit also from info dealing with O layer thickness changes through time since fire. It would be good also to present the Cslow and Cfast values for different soil horizons. Right now there are two separate paragraphs in discussion dealing with soil carbon bioreactivity (separately for FH horizon and mineral soil), but in results section one can't find the values for these two horizons separately (see my detailed comments under Results and Discussion), instead authors are presenting the combined values (Fig. 3). And this brings us to another problem – authors are stating (in discussion) that 73% of the C in FH horizon is acid-insoluble. However, is not shown in results section, nor discussed in discussion section, that are there differences in recalcitrance of the soil in FH horizon (this is the part of the soil that is most affected by fire) through time since fire.

Below are my detailed comments on the manuscript: Abstract P1 L16: Here and later in the text, if you started to use abbreviations "carbon (C)", please be constant through the text.

Introduction P1 L33: Here and later in the text, if you started to use abbreviations "carbon (C)", please be constant through the text. Change "carbon-cycle" to "C-cycle". Later in the text also change "carbon-quality", carbon balance", etc. P2 L76: If you have started to use "FH horizon" (actually we are missing explanation for that), why to

jump her into "O layer"? It occurs also later in text.

Material and Methods P3 L93: Definition/explanation for FH horizon is needed. You are also using "O layer" in text, that is maybe more understandable for the reder P3 L96-98: These 400 cm2 microplots for moss biomass measurements, how many of them per sample plot? P3 L114-117: What about mineral soil from 15-35 cm depth? I can understand that you were missing that sample from one plot, but here the soil preparation of samples from that depth is not described at all. P3 L120: What do you mean with "B-horizon" here? You haven't been describing the soil horizons. Are these now samples from mineral soil from depth 0-15 cm, or 16-35 cm? P4 L137-140: Incubation temperature +26°C? Why so high temperature? The chosen incubation temperature is not representing anyhow the conditions (soil temperature) in the field. Usually during the incubation the temperature is chosen to be similar to the field conditions, and also different temperatures are used. Why in this study the samples were incubated with only one temperature? The respiration rates increase rapidly with higher temperatures, and if using much higher temperatures (soil temperatures) than one can find from the field, the outcomes could be unexpected. P4 L148-151: It would be good to know how the CO2 measurement times (sample taking times after closure) changed due to soil layer and progress of the experiment. P5 L157-165: As the incubation was done with only one (really high) temperature, I think the standardization to 24 hour period and the cumulative C mineralization calculations can be really biased, as the temperature is not taken into account.

Results and Discussion P8 L193-301: Would it be possible to see the "Cslow" and "Cfast" also for different soil horizons through TSF? I also can't find from the manuscript the O layer or FH horizon thickness changes through TSF. P10 L364-365: Is the talk now about completely insoluble C, or this actually includes also acid- soluble C in FH horizon? Is this 73% now some kind of average for entire 314 year chronosequence? What are the values close to the fire and through succession? There have been some studies from Northern-America lately, dealing with forest fires and soil organic matter

quality in permafrost soils. It would be interesting actually to compare the findings. P11 L391: It would be interesting to see the values of insoluble/acid soluble C (%) for mineral soil in this section (as was presented in previous section for FH horizon) and this for entire chronosequence. P11 L256-260: To long and confusing sentence, consider rephrasing. There is also a lot of talk considering Figure S2 from supplementary material. If this figure is so important, why to include it into the supplementary material? Conclusions Conclusion should be short summary of your work and findings. Currently there are many other studies (with references) included into summary, that actualy should belong into the discussion section

Figures Like mentioned earlier, it would be good to see the "Cslow" and "Cfast" also for different soil horizons through TSF (in separate figure) Tables Table 1. Min and Max of FH depth, soil thickness, pH Bulk density particle size is not giving much to the readers as it is not known on what side of the succession these values are (close to the fire or at the end of succession and chronosequence). It would be much more informative to give these values through chronosequences (starting close to fire and then with certain interval after the fire)

---

## Referee Comment (RC2) · Anonymous Referee #2 · 11 Feb 2020

In this manuscript, the authors use 1-year soil incubations to analyze how "bioreactive" and "recalcitrant" soil C pools vary over a 350 year fire chronosequence. They use a linear regressions and confirmatory path analysis to test hypothesized cause-and-effect relationships between the soil pools and other soil chemistry variables. In general, the manuscript is informative, well-supported, and easy to read.

General comments:

1. I would like to see the correlation between incubation-derived estimate of Cslow and the acid-insoluble residue as a figure, since the latter is often used as a proxy for the former without direct comparison. In this manuscript, it's not clear which estimate of

bioreactive/recalcitrant C is used in the models and in the figures.

2. Why is soil texture hypothesized to influence moss dominance in the causal models?

3. Moss community composition (Sphagnum vs feather moss) is included, but is there any difference in moss abundance that could be included in the model? Even non-sphagnum mosses have distinct biochemistry and low decomposition rates compared to vascular plants.

4. The conclusion contains lots of new analysis not included elsewhere in the manuscript and would be better presented as an additional discussion section

5. Discussion of temperature generates some confusion about what is actually being measured. Since the incubation temperature (26°C) is much warmer than the MAT of the study site, the bioavailability assays correspond to "potentially available" C more than a realistic estimate of in situ soil respiration. This is a reasonable choice but leads to some confusion in the introduction and discussion:

-The value of analyzing the recalcitrant SOC fraction is justified in regards to the C-quality temperature hypothesis, since "recalcitrant" C should respond more to warming. But, since the incubation temperature is 26C, the "recalcitrant" C that is actually measured is SOC that is preserved even when temperature is increased to unrealistically high levels.

-Lines 378-383 seems to suggest that climate does not drive SOM decay rates or transfer between SOC pools, which is not supported by the study.

-The connection between these results and the temperature sensitivity of soil respiration (lines 385-390) is also unclear. This section could be improved by discussing the relationship between hypothesis FH1 and the C-quality temperature hypothesis, and the implications of the results for the validity of the CQTH.

---

## Author Response (AR1)

Response to RC1

We wish to thank reviewers for dedicating their time to review our manuscript and for their comments that are very helpful to improve our manuscript. We address all the comments raised by Anonymous Referee #1 (in bold) below.

General comments In this study, the authors were evaluating post fire carbon stock changes in functional reservoirs (bioreactive and recalcitrant) using the proportion of C mineralized in CO2 by microbes in a long-term lab incubation, as well as the proportion of C resistant to acid hydrolysis. Through the manuscript (already in Abstract) there are problems with abbreviations, one can find through the text carbon and C, bioradiactive C and CBioR, carbon dioxide and CO2, etc. If you have started to use abbreviations, please be constant through entire text.

We acknowledge that there are some problems with abbreviations. We will solve all the abbreviation problems in the manuscript to be constant throughout the manuscript, by doing the following modifications:

L. 16, 18, 20, 26, 33, 36, 40, 379, 418, the word "carbon" will be abbreviated with "C" but it will remain not abbreviated in the titles.

L. 18, 20, "Mn" and "Al" will be changed for full words "manganese" and "aluminum", respectively.

L. 63, "CBioR" will be replaced by "bioreactive C stock".

L. 65, because they are used for the first time in the manuscript (abstract excluded) we will add the significance for "Mn" and "Al" abbreviations.

L. 73-75 will be replaced by "[...] 1) soil  $C_{BioR}$  increases as forest stands get older, leading to a buildup of soil bioreactive C stock under the cold conditions of the boreal forest; 2) alternatively, if the bioreactive soil C stock reaches a new equilibrium because of rapid turnover, the proportion of bioreactive C stock should decline as total soil C stock increases with TSF".

L. 144, 152: according to the Soil journal manuscript preparation guidelines and because it is not ambiguous, we will use "CO2" instead of "Carbon dioxide".

L. 209, we will remove "manganese (Mn)" to keep only "Mn".

L. 262, "CBioR" will be replaced by "C lability".

L. 294, "CBioR" will be replaced by "bioreactive C".

L. 342, we will replace "exchangeable aluminum" by "exchangeable Al".

L. 425, "time since fire" will be replaced by "TSF".

Introduction is informative, but I would expect more talk on the topic, why the C bioreactivity is important, and what does it mean if we have the changes in C bioreactivity reservoirs through fire chronosequences. According to Anonymous Referee #1, we have not fully introduced the usefulness of better understanding soil C bioreactivity in the section *1 Introduction*. Several improvements will be done, as follow:

L. 39: "[...] on soil heterotrophic respiration. Furthermore, the bioreactive C fraction of soil organic matter is cycled on time scales relevant to global warming. Thus, quantifying the size of the bioreactive soil C reservoir and understanding the controlling factors of soil C bioreactivity  $(C_{BioR})$  is key to inform models of C cycle in face of and to better anticipate global warming". L. 47: "[...] from the decomposition of soil organic matter by clay surfaces (Six et al., 2002). At steady state and when the accumulation of physico-chemically stabilized soil C occurs, soil C can only accumulate as non-protected C (Castellano et al., 2015) that is more prone to decompose quickly".

L. 55: "[...] a relative measure of soil C lability (Laganière et al., 2015; Xu et al., 1997). Assessing the sizes of resistant and bioreactive soil C fractions through a fire chronosequence would help modelers to enhance the current and future C balance of landscape prone to wildfires".

**The hypothesis at the end of the introduction are OK, but when the other set of hypothesis are presented in Material and methods section, this creates some confusion.**

The hypotheses at the end of the introduction are general hypotheses, whereas the hypotheses included in the second section *2 Material and methods* include detailed hypotheses of the third general hypothesis. To clarify this point, we will modify the text as follows: L. 72-77 will be replaced by (also including the modifications mentioned previously) "From there, our general hypotheses are that once site factors such as overstory composition, surficial deposits and soil drainage are accounted for, as they were in the present study: 1) soil CBioR increases as forest stands get older, leading to a buildup of soil bioreactive C stock under the cold conditions of the boreal forest; 2) alternatively, if the bioreactive soil C stock reaches a new equilibrium because of rapid turnover, the proportion of bioreactive C stock should decline as total soil C stock increases with TSF; and 3) soil CBioR is primarily controlled by TSF and moss dominance in the O layer (FH horizon), and by soil physico-chemistry in the mineral soil (see 2.3 for detailed hypotheses)."

L. 181 will begin with "According to our third general hypothesis and to address the complex interplay among climatic and non climatic factors, [...]".

**Study design needs some improvements (see my detailed comments), as currently it is difficult to understand how many microplots (for moss biomass measurements) were established per sample plot (maybe a scheme describing the measurements from sample plot would be useful as supplementary material).**

All the details about the field work can be found in the section 2.1 Site selection, sampling design and field work. We have explained on L. 95-98 that for each plot we used 20 microplots to

identify the dominant moss types, at the same place where we measured the thickness of the FH horizon. According to Anonymous Referee #1, a summary of the field work would help the readers, so we will add a diagram of the sampling plot as supplementary materials (as Fig. S1).

**I also can't understand why the samples were incubated with so high temperature (+26â `D `C), and only with one temperature. Usually, during the incubation, one attempts to mimic the field conditions (use temperatures similar to real soil temperature). And due to that I'm really concerned that are the cumulative respiration calculations actually valid.**

We thank Anonymous Referee #1 for this comment but there is a misunderstanding here as our goal was not to mimic real soil temperature as observed in the field. Our goal with the incubation was to assess the accessible soil C for microbial decomposition that is the functional soil C reservoir contributing to greenhouse gas emission. In our study, we chose to incubate soil samples at 26°C according to the literature (temperatures between > 0°C and 35°C are the most commonly used). Low C mineralization rates were observed for boreal forest soils incubated under 15°C (see Paré et al. 2011; Laganière et al. 2015). So 26°C is a trade-off to maximize CO2 production by the microcosms while performing measurements in the optimum temperature range for the microbial breakdown of soil organic matter and this has been verified globally (Carey et al. 2016). We were mostly interested in the soil C bioreactivity, that is why we have incubated soil samples with only one temperature. Usually, several temperatures for soil incubation are used to assess the Q10 value, i.e., a relative measure of temperature sensitivity of C mineralization (Laganière et al. 2015). Due to the high number of soil samples implying a tedious logistic for the incubation experiment, we did not use several temperatures to assess the Q10 index.

**Results and Discussion section could benefit also from info dealing with O layer thickness changes through time since fire. It would be good also to present the Cslow and Cfast values for different soil horizons.**

The topic of O layer thickness and soil organic matter accumulation was studied previously and details about some O layer properties (as O layer thickness, bulk density and C concentration) can be found in a previous study (Andrieux et al., 2018). We acknowledge that presenting  $C_{slow}$  and  $C_{fast}$  values for the different soil layers would help the readers to have a full picture of post-fire changes occurring with soil depth. Based on Anonymous Referee #1 comments, we will include more details about the O layer thickness in Table 1, for several post-fire age-classes. Moreover, we will add: 1) a figure in the supplementary material (soil C quality by pool and for each soil layer as a function of time since fire); 2) the accumulation rate for each of the soil layers in Table 2 and; 3) we will also enhance the sections *3.1 Post-fire soil C pool size* and *4.1 Post-fire soil C quality*, as follows:

L. 301, integrating a new paragraph: "These general trends are mostly influenced by the size of the  $C_{slow}$  and  $C_{fast}$  pools of the O layer, being 5 times and 2.4 times larger than the top 35 cm of the mineral soil one (i.e., sum of the two mineral soil layers), respectively (Table 2; Fig.Sx).  $C_{slow}$

and  $C_{fast}$  decrease with soil layers from the surface soil horizon (O layer) to the deeper mineral soil, both in absolute size and proportion (Table 2 ; Fig. Sx). Consistently with the whole data set, the proportion of  $C_{slow}$  and  $C_{fast}$  do not vary quantitatively with TSF for all the soil layers analyzed separately (Table 2). The size of the  $C_{slow}$  pool increases linearly with TSF in the O layer only ( $R^2 = 0.09$ , p = 0.01), not in mineral soil layers (p > 0.07 for both mineral soil layers). The size of the  $C_{fast}$  pool increases linearly with TSF in the O layer ( $R^2 = 0.12$ , p = 0.003) and in the top 15 cm of the mineral soil ( $R^2 = 0.05$ , p < 0.05), not in the deepest mineral soil layer from 15 to 35 cm (p > 0.21)."

L. 356: "[...], such as cold temperatures under a thickening O layer developed with TSF (Table 1), could have slowed down labile C degradation and allow its accumulation (Kane et al., 2005). Our results also emphasize that changes in the size of the soil functional reservoirs with TSF are stratified within the soil profile. This pattern is consistent with the fact that fire impact on soil C stock is limited to surface soil horizons (Andrieux et al., 2018)".

Right now there are two separate paragraphs in discussion dealing with soil carbon bioreactivity (separately for FH horizon and mineral soil), but in results section one can't find the values for these two horizons separately (see my detailed comments under Results and Discussion), instead authors are presenting the combined values (Fig. 3). And this brings us to another problem – authors are stating (in discussion) that 73% of the C in FH horizon is acid-insoluble. However, is not shown in results section, nor discussed in discussion section, that are there differences in recalcitrance of the soil in FH horizon (this is the part of the soil that is most affected by fire) through time since fire.

More details about soil C pool sizes will be added separately for the FH horizon and the mineral soil layers (please, refer to our previous response).

**Below are my detailed comments on the manuscript: Abstract P1 L16: Here and later in the text, if you started to use abbreviations "carbon (C)", please be constant through the text.**

All the abbreviation problems will be solved (please, see our previous response).

**Introduction P1 L33: Here and later in the text, if you started to use abbreviations "carbon (C)", please be constant through the text. Change "carbon-cycle" to "C-cycle". Later in the text also change "carbon-quality", carbon balance", etc.**

All the abbreviation problems will be solved (please, see our previous response).

**P2 L76: If you have started to use "FH horizon" (actually we are missing explanation for that), why to jump her into "O layer"? It occurs also later in text. Material and Methods P3 L93: Definition/explanation for FH horizon is needed. You are also using "O layer" in text, that is maybe more understandable for the reder**

We acknowledge that using "O layer" instead of "FH horizon" will be more understandable for the reader, so we will replace "FH horizon" with "O layer" through the text (L. 20, 26, 76, 97, 98, 101, 114, 115, 118, 128, 133, 145, 150, 187, 199, 212, 241, 253, 259, 277, 284, 285, 293, 302, 304, 306, 311, 320, 322, 326, 356, 359, 361, 362, 363, 365, 370, 372, 374, 375, 384, 392, 401, 410, 442) and in tables, figure and figure captions. Moreover, the hypotheses named "FH1" and "FH2" will be renamed "O1" and "O2" (L. 193, 197, 199, 212, 213, 214, 219, 220, 228, 303, 305, 308, 319).

**P3 L96-98:** These 400 cm2 microplots for moss biomass measurements, how many of them per sample plot?**

We used 20 microplots per plot, as mentioned L. 96 (please, see also our previous response).

**P3 L114-117: What about mineral soil from 15-35 cm depth? I can understand that you were missing that sample from one plot, but here the soil preparation of samples from that depth is not described at all.**

All the mineral soil samples (0-15 cm, 15-35 cm and first 15 cm of the B horizon) were prepared following the same standard protocol. We will add this precision as follows: L. 115-117 (including previous corrections mentioned above) "O layer samples were sieved through a 6-mm mesh before being oven-dried (60°C), whereas mineral soil samples (either top 15 cm of the mineral soil, mineral soil from 15 to 35 cm or top 15 cm of the B horizon) were dried by air and passed through a 2-mm sieve".

**P3 L120: What do you mean with "B-horizon" here? You haven't been describing the soil horizons. Are these now samples from mineral soil from depth 0-15 cm, or 16-35 cm?**

Pyrophosphate extractable Fe and Al analyses were done for the top 15 cm B horizon samples only. We will clarify this point as follows:

L. 119-121 : "Finely ground sub-samples (< 0.5 mm) were used for C concentration, pyrophosphate extractable Fe and Al (top 15 cm B horizon only) and acid hydrolysis analyses." B horizon pyrophosphate extractable Fe and Al are used to determine if a soil is classified as a podzol or not and to define to which sub-group it belongs to in the Canadian System of Soil Classification. Moreover, the characteristics of the B horizon mirrors the soil processes of podzolization (Shaetzl and Anderson, 2005). In our study, we used the pyrophosphate extractable Fe and Al (i.e., metal oxides) of the B horizon as a proxy for the podzolisation status of the soil. Also, we will add a short description of the soil horizons and redirect the reader to the diagram of the plot inventory and soil sampling were we will add a photo of a soil profile (see our previous response), as follows:

L. 84: "The soils that develop under this cool and humid climate with acidic litter inputs typically belong to the Podzolic order (Table 1). In boreal forests, podzolic soils often have a thick organic surface horizon (O layer), an eluviated A horizon (Ae) from where leached materials accumulate in the illuviated B horizon (Fig. Sx)".

P4 L137-140: Incubation temperature +26â °D °C? Why so high temperature? The chosen incubation temperature is not representing anyhow the conditions (soil temperature) in the field. Usually during the incubation the temperature is chosen to be similar to the field conditions, and also different temperatures are used. Why in this study the samples were incubated with only one temperature? The respiration rates increase rapidly with higher temperatures, and if using much higher temperatures (soil temperatures) than one can find from the field, the outcomes could be unexpected.

Our aim was not to assess the decomposition rate of soil organic matter that could be observed *in situ*. The goal of our experiments was to assess the accessible soil C for microbial decomposition, that is the functional soil C reservoir contributing to greenhouse gas emission. For this purpose, long-term lab incubation at high temperature have most often been used (Paul et al., 2006). We used a consistent method for all our soil samples, so we are confident to compare our results among all the studied sites, including the bioreactivity of soil C that is a relative measure of soil C lability. A global analysis also revealed that soil heterotrophic respiration was optimal around this temperature (Carey et al. 2016). But we do agree with Anonymous Referee #1 that our results do not allow extrapolations of C mineralization rate that could be observed in the field. Please, see also our previous response.

**P4 L148-151: It would be good to know how the CO2 measurement times (sample taking times after closure) changed due to soil layer and progress of the experiment.**

Overall, final  $CO_2$  measurements were done 4h and 24h after sealing the jars for O layer samples and mineral soil samples, respectively, because O layer  $CO_2$  production rate was greater. Some exceptions have been made. Nevertheless, this sample taking time after closure is meaningless because all  $CO_2$  production rate were standardized on a daily (24 h) basis.

P5 L157-165: As the incubation was done with only one (really high) temperature, I think the standardization to 24 hour period and the cumulative C mineralization calculations can be really biased, as the temperature is not taken into account.

Rescaling the respiration rate on a 24 h basis is inherent to the soil incubation method (Paré et al., 2006) and allow the direct comparison among samples. The temperature remained constant through the course of the experiments (not only during measuring periods), so this parameter did not have affected our estimations. Please, see also our previous responses.

**Results and Discussion**

**P8 L193-301: Would it be possible to see the "Cslow" and "Cfast" also for different soil horizons through TSF? I also can't find from the manuscript the O layer or FH horizon thickness changes through TSF.**

Here, we assume that Anonymous Referee #1 comments on L. 293-301. As mentioned above, we will add more details about soil C pool sizes separately for the O layer and the mineral soil layers (please, refer to our previous response).

P10 L364-365: Is the talk now about completely insoluble C, or this actually includes also acid- soluble C in FH horizon? Is this 73% now some kind of average for entire 314 year chronosequence? What are the values close to the fire and through succession? There have been some studies from Northern-America lately, dealing with forest fires and soil organic matter quality in permafrost soils. It would be interesting actually to compare the findings.

As shown in the subtitle 4.2.1 Soil carbon bioreactivity in the FH horizon, here we deal with the O layer only. Moreover, the sentence L. 364-365 we clarified the sentence as mentioned above: "[...] the high proportion of acid-insoluble C of O layer samples ( $73 \pm 5\%$ , data not shown)". We acknowledge that it was not clear that 73% was the average value for all the O layer samples. We will clarify as follows:

L. 364-365: "This is reflected in the high proportion of acid-insoluble C of the O layer samples (among all the O layer samples, mean  $\pm$  sd = 73  $\pm$  5%; Table 1)". Here, we will refer the reader to Table 1 in which we will give the proportion of acid-insoluble soil C sorted in several classes of time since fire.

P11 L391: It would be interesting to see the values of insoluble/acid soluble C (%) for mineral soil in this section (as was presented in previous section for FH horizon) and this for entire chronosequence.

According to Anonymous Referee #1 previous comment, values of acid-insoluble C by proportion will be included in the manuscript. Please, see our previous response.

**P11 L256-260: To long and confusing sentence, consider rephrasing.**

There is a mismatch between P. 11 and L. 256-260. L. 256-260 explains the terms of the equations (4) and (5). We are sorry not to be able to respond to this comment.

**There is also a lot of talk considering Figure S2 from supplementary material. If this figure is so important, why to include it into the supplementary material?**

Fig. S1 (there is only one figure in supplements) represents all the  $CO_2$  measurements that we have done during the course of the incubation experiment. We have referred only twice to this figure in the manuscript (L. 146 and 158). We wanted to show that the shapes of the curves are consistent with the ones often observed with the incubation method of soil samples. However, because our paper does not focus on methodological issues of soil incubation, we would prefer to keep Fig. S1 in supplements.

**Conclusions**

Conclusion should be short summary of your work and findings. Currently there are many other studies (with references) included into summary, that actualy should belong into the discussion section

According to Anonymous Referee #1, we acknowledge that the current conclusion section contains information that should be included in the discussion section. Therefore, this section will be renamed *4.2.3 Implication for C cycling and research needs*.

**Figures**

**Like mentioned earlier, it would be good to see the "Cslow" and "Cfast" also for different soil horizons through TSF (in separate figure)**

A separate figure will be added. Please, see our previous response.

**Tables Table 1.**

Min and Max of FH depth, soil thickness, pH Bulk density particle size is not giving much to the readers as it is not known on what side of the succession these values are (close to the fire or at the end of succession and chronosequence). It would be much more informative to give these values through chronosequences (starting close to fire and then with certain interval after the fire)

The minimum, maximum, mean and standard deviation of values included in Table 1 are informative because the readers can have a look at the variability of both response and explanatory variables used in the subsequent analyses. According to Anonymous Referee #1,

giving these values through the chronosequence is relevant, so we will add this information for all post-fire age-classes. Please, see also our previous responses.

Response to RC2

We wish to thank the reviewers for dedicating their time to review our manuscript and for their comments that are very helpful to improve our manuscript. We address all the comments raised by Anonymous Referee #2 (in bold) below.

In this manuscript, the authors use 1-year soil incubations to analyze how "bioreactive" and "recalcitrant" soil C pools vary over a 350 year fire chronosequence. They use a linear regressions and confirmatory path analysis to test hypothesized cause and-effect relationships between the soil pools and other soil chemistry variables. In general, the manuscript is informative, well-supported, and easy to read.

We thank Anonymous Referee #2 for her/his positive comments.

**General comments:**

**1.** I would like to see the correlation between incubation-derived estimate of Cslow and the acidinsoluble residue as a figure, since the latter is often used as a proxy for the former without direct comparison. In this manuscript, it's not clear which estimate of bioreactive/recalcitrant C is used in the models and in the figures.

In our study, we did not use incubation results to derive estimates of  $C_{slow}$ .  $C_{slow}$  was estimated as the fraction that is acid-insoluble. Nevertheless, incubated-derived  $C_{slow}$  can be calculated as the residuals of incubated-derived  $C_{fast}$  substracted to total C. So, the correlation between  $C_{BioR}$  and acid-insoluble residues would inform if acid hydrolysis and incubation of soil samples methods give consistent results to estimate soil C cycling. We will add a figure with these correlations in supplements and a short paragraph in the discussion section.

We agree in part with Anonymous Referee #2 about clarifying which estimate of bioreactive or recalcitrant C is used in the models. Indeed, the response variable at the center of the direct acyclic graph in Fig.2, Fig.4 and Fig.5 is CBioR. CBioR has been defined in the section *1 Introduction* "as the proportion of C mineralized in CO2 by microbes at constant temperature and constant water content over a long period of times as a relative measure of soil C lability (Laganière et al., 2015; Xu et al., 1997)" (L. 54-55). Moreover, we explain in the section *2.4.2 Soil C quality and bioreactivity* (L. 252-264) that Cfast and Cslow are functional reservoir sizes (so stocks) calculated from the incubation experiment and from the acid hydrolysis experiment, respectively, and that we have analyzed these response variables as a function of TSF (referring to Fig.3 in which legend and caption use the same abbreviations, see also L. 266-267). However, we will bring more clarity in the text, as follows:

L. 51-52 (inputs): "As part of this study, we characterized the acid-insoluble and bioreactive soil organic C pools ( $C_{slow}$  and  $C_{fast}$ , respectively, expressed as stocks) that accumulate following fire."

L. 252: "First, we wanted to estimate variation in the size of the bioreactive or recalcitrant soil C pools ( $C_{fast}$  and  $C_{slow}$ , respectively, expressed as stocks) with TSF."

**2. Why is soil texture hypothesized to influence moss dominance in the causal models?**

We acknowledge that the explanation is lacking. We will add a short sentence, as follows:

L. 220: "[...] their influence on moss dominance. Indeed, we expected that Sphagnum spp. would dominate over feathermosses under wetter conditions induced by greater precipitation, fined-texture soils holding more water, or both, because of their greater dependence to high soil water content."

**3.** Moss community composition (Sphagnum vs feather moss) is included, but is there any difference in moss abundance that could be included in the model? Even non-sphagnum mosses have distinct biochemistry and low decomposition rates compared to vascular plants.**

In this study, we included an index of moss dominance only based on a presence/absence survey (section 2.4.1 Index of moss dominance, L. 240-250). Neither moss community composition nor moss abundance was sampled for this study.

**4. The conclusion contains lots of new analysis not included elsewhere in the manuscript and would be better presented as an additional discussion section**

According to both Anonymous Referee #1 and Anonymous Referee #2, we will transfer the section 5 *Conclusion* to the discussion section. Please, see also our response to Anonymous Referee #1.

5. Discussion of temperature generates some confusion about what is actually being measured. Since the incubation temperature  $(26 \circ C)$  is much warmer than the MAT of the study site, the bioavailability assays correspond to "potentially available" C more than a realistic estimate of in situ soil respiration. This is a reasonable choice but leads to some confusion in the introduction and discussion:

-The value of analyzing the recalcitrant SOC fraction is justified in regards to the C-quality temperature hypothesis, since "recalcitrant" C should respond more to warming. But, since the incubation temperature is 26C, the "recalcitrant" C that is actually measured is SOC that is preserved even when temperature is increased to unrealistically high levels.

-Lines 378-383 seems to suggest that climate does not drive SOM decay rates or

transfer between SOC pools, which is not supported by the study.

-The connection between these results and the temperature sensitivity of soil respiration (lines 385-390) is also unclear. This section could be improved by discussing the relationship between hypothesis FH1 and the C-quality temperature hypothesis, and the implications of the results for the validity of the CQTH.

- In our study, the "recalcitrant" soil C was assessed with acid-hydrolysis of soil samples. This method assumes that the non-hydrolysable C fraction is not accessible for microbial degradation (L. 52-53; L. 167-168). Nevertheless, this point of view has practical implications for modeling purposes (Paul et al., 2006) but is controversial (Kleber, 2010; Lehman and Kleber, 2015). That's why we make a strong case on CBioR and not on "recalcitrant" C. Indeed, in our study CBioR correspond to "potentially available" C (i.e., relative measure of C lability under standard conditions), our data has to remain in the context of lab incubation, and cannot be extrapolated to *in situ* soil respiration. All our samples have been processed equally, so the results of soil incubation (CBioR) are comparable among our soil samples and highlight some of the processes driving the potential C loss from boreal soils through microbial respiration. Moreover, we have indicated that the "recalcitrant" C can be processed by microbes synthesizing enzymes involving Mn (L. 363-371).

- To avoid confusions, we will add "also" into L. 382, as follows:

L. 382: "[...] such as exchangeable Mn concentrations and pH, might also be used to modulate soil C dynamics in such models, [...]"

- Because we have not incubated our soil samples at several temperatures, we cannot assess the temperature sensitivity of soil respiration (Q10, please see our response to Anonymous Referee #1). So, we do not feel comfortable discussing the CQTH hypothesis. Nevertheless, we acknowledge that L. 385-390 is unclear, so we will modify it as follows:

L. 385-387: "Furthermore, when synthesizing data of *in situ* experimental warming, Carey et al. (2016) found no change in soil respiration rate for warmed compared to control plots at the global scale, whereas changes were found to be significant for the boreal biome."

**List of relevant changes**

All the modifications that we have proposed in our responses to reviewers have been included. The relevant changes are listed below:

- We have harmonized the abbreviations through the manuscript;
- We have developed the results in the section *3.1 Post-fire soil C pool size* by adding a description of the changes in soil C functional pool separately for all soil layers (O layer, mineral soil 0-15 cm and mineral soil 15-35 cm);
- We supplemented the discussion with the section *4.3 Implication for carbon cycling and research needs* with the text that was part of the conclusion in the early version of the manuscript;
- We add a conclusion including summary of the key outcomes and implications of the study only;
- Some information about general characteristics of the sampling sites have been detailed by ageclass in Table 1;
- We have added post-fire soil C pool size and accumulation rate for each of the soil layer in Table 2;
- We have added a diagram of the sampling plot in supplements (Fig. S1);
- We have added the carbon quality data (stock and fraction of total stock) for each of the soil layer in supplements (Fig. S3).

**Boreal forest soil chemistry drives soil organic carbon bioreactivity along a 314-year fire chronosequence**

Benjamin Andrieux1, David Paré2, Julien Beguin2, Pierre Grondin3, Yves Bergeron1

5 1NSERC-UQAT-UQAM Industrial Chair in Sustainable Forest Management, Forest Research Institute, Université du Québec en Abitibi-Témiscamingue, Rouyn-Noranda, QC, J9X5E4, Canada 2Natural Resources Canada, Canadian Forest Service, Laurentian Forestry Centre, Québec, QC, G1V4C7, Canada 3Ministère des Forêts, de la Faune et des Parcs, Direction de la Recherche Forestière, Québec, G1P3W8, Canada *Correspondence to*: Benjamin Andrieux (benjamin.andrieux@uqat.ca)

- 10 Abstract. Following wildfire, organic carbon (C) accumulates in boreal forest soils. The long-term patterns of accumulation as well as the mechanisms responsible for continuous soil C stabilization or sequestration are poorly known. We evaluated post-fire C stock changes in functional reservoirs (bioreactive and recalcitrant) using the proportion of C mineralized in CO2 by microbes in a long-term lab incubation, as well as the proportion of C resistant to acid hydrolysis. We found that all soil C pools increased linearly with time since fire. The bioreactive and acid-insoluble soil C pools increased at a rate of 0.02 MgC.har
- 15 1.yr-1 and 0.12 MgC.ha-1.yr-1, respectively, and their proportions relative to total soil C stock remained constant with time since fire (8% and 46%, respectively). We quantified direct and indirect causal relationships among variables and C bioreactivity to disentangle the relative contribution of climate, moss dominance, soil particle size distribution and soil chemical properties (pH, exchangeable manganese and aluminum, and metal oxides) to the variation structure of *in vitro* soil C bioreactivity. Our analyses showed that the chemical properties of Podzolic soils that characterise the study area were the best predictors of soil
- 20 C bioreactivity. For the O Jayer, pH and exchangeable manganese were the most important (model-averaged estimator for both: 0.34) factors directly related to soil organic C bioreactivity, followed by time since fire (0.24), moss dominance (0.08) and climate and texture (0 for both). For the mineral soil, exchangeable aluminum was the most important factor (model-averaged estimator: -0.32), followed by metal oxide (-0.27), pH (-0.25), time since fire (0.05), climate and texture (~ 0 for both). Of the four climate factors examined in this study (i.e., mean annual temperature, growing degree-days above 5°C, mean
- 25 annual precipitation and water balance) only those related to water availability, and not to temperature, had indirect effect ( layer) or a marginal indirect effect (mineral soil) on soil ( bioreactivity. Given that predictions of the impact of climate change on soil c balance are strongly linked to the size and the bioreactivity of soil C pools, our study stresses the need to include the direct effects of soil chemistry and the indirect effects of climate and soil texture on soil organic matter decomposition in Earth System Models to forecast the response of boreal soils to global warming.

**30 1 Introduction**

35

Soil is the largest terrestrial carbon (C) reservoir (Scharlemann et al., 2014) and a major source of uncertainty in ecosystem C predictions (Shaw et al., 2014). Therefore, an advanced mechanistic understanding of soil C processes needs to be investigated and integrated into forecast models to reduce uncertainties in global *C*-cycle feedback projections and to better predict the effects of global change on soil C reservoir (Bradford et al., 2016; Schmidt et al., 2011). The maintenance of the vast soil C reservoir is partly under microbial control (Cotrufo et al., 2013) and could respond to variations in environmental conditions (Davidson and Janssens, 2006). Hence, the *C*-quality temperature hypothesis states that more "recalcitrant" soil organic matter should have higher temperature sensitivity (Craine et al., 2010; Fierer et al., 2005). According to this hypothesis, it is important to distinguish the recalcitrant portion of the soil organic matter from the active portion in order to predict the impact of a rise in temperature on soil heterotrophic respiration. *Furthermore, the bioreactive C fraction of soil organic matter is cycled on*

|------------------------|
|                        |

| _ |                                               |
|---|-----------------------------------------------|
|   |                                               |
|   |                                               |
|   |                                               |
| _ |                                               |
|   |                                               |
|   |                                               |
| - | Ais en forme · Police ·10 pt Anglais (Canada) |

- 55 time scales relevant to global warming. Thus, quantifying the size of the bioreactive soil C reservoir and understanding the controlling factors of soil C bioreactivity (CBioR) is key to inform models of C cycle in face of and to better anticipate global warming.
- Wildfire is a major natural disturbance in boreal forests that drives the ecosystem C balance (Bond-Lamberty et al., 2007; Kurz et al., 2013) and is known to impact several soil properties, including organic matter quantity and quality (Certini, 2005;
  Knicker, 2007). Key soil properties, some evolving following fire (e.g., soil acidity) and some not (e.g., particle size distribution), interact with climate and vegetation composition in complex causal direct and indirect relationships to regulate post-fire soil C accumulation (Andrieux et al., 2018). A saturation of soil C accumulation, especially for its recalcitrant portion, is often observed in soils when the rates of organic matter input to the soil are increased (Stewart et al., 2007; Hassink, 1996). Saturation of recalcitrant C is believed to come from the finite capacity of stabilization mechanisms in soils, such as chemical
- 65 protection from the decomposition of soil organic matter by clay surfaces (Six et al., 2002). At steady state and when the accumulation of physico-chemically stabilized soil C occurs, soil C can only accumulate as non-protected C (Castellano et al., 2015) that is more prone to decompose quickly. However, the long-term patterns of change in soil C quality and the accumulation pattern of recalcitrant and bioreactive C pools as well as the mechanisms responsible for continuous accumulation or stabilization of soil C reservoirs are poorly known and have not been explicitly integrated into soil
- 70 biogeochemistry (Luo et al., 2016). Most models of soil C dynamics divide soil organic matter into several conceptual pools and simulate decomposition as a first-order decay process (Luo et al., 2016). As part of this study, we characterized the acid-insoluble and bioreactive soil organic C pools (Cglow and Cfast, respectively, expressed as stocks) that accumulate following wildfire. The acid-insoluble soil C fraction is assumed to be "recalcitrant" or resistant to biological degradation (Paul et al., 2006; Xu et al., 1997). Hereafter, we define CBioR as the proportion of C mineralized in CO2 by microbes at constant
- 75 temperature and constant water content over a long period of time as a relative measure of soil C lability (Laganière et al., 2015; Xu et al., 1997). Assessing the sizes of resistant and bioreactive soil C fractions through a fire chronosequence would help modelers to enhance the current and future C balance of landscape prone to wildfires.

[revised manuscript text omitted]

matter quality (Johnson and Curtis, 2001), resulting in a decrease of  $C_{BioR}$  with increasing TSF. Mn availability could directly modulate  $C_{BioR}$  (see Q1), and exchangeable Al could impede microbial decomposition when in excess (Kunito et al., 2016).

**2.3.4 Alternative hypothesis for the mineral soil, MIN2**

As an alternative to hypothesis MIN1, this hypothesis assumes that only TSF, pH, metal oxides and Mn/Al have direct effects on  $C_{BioR}$ . Additionally, pH is assumed to decrease with TSF because of the imbalance in nutrient uptake caused by aggrading vegetation. Also, exchangeable cations are dependent on pH (Sanborn et al., 2011), and the decrease in pH favours the creation of organometallic complexes impeding microbial decomposition (Buurman and Jongmans, 2005; Porras et al., 2017). Contrary to hypothesis MIN1, which assumed direct effects of climate and soil texture on  $C_{BioR}$ , this hypothesis assumes that climate and soil texture have only indirect effects on  $C_{BioR}$ . The indirect effect of climate on  $C_{BioR}$  is mediated through its effect on mineral weathering (Doetterl et al., 2015) and the quantity of metal oxides leached from the upper soil layers (Schaetzl and Anderson, 2005). Compared to coarse-textured soils, fine-textured soils have more reactive surface sites that can bind

additional Mn and Al ions (Petersen et al., 1996).

**2.4 Calculations and data analyses**

**2.4.1 Index of moss dominance**

300

305

In order to account for the effects of moss functional traits on CBioR of the O layer, we differentiated between *Sphagnum* spp. 310 and feather mosses, since they have different ecophysiological characteristics (Bisbee et al., 2001), e.g., feather mosses decompose faster than Sphagnum spp. (Fenton et al., 2010; Lang et al., 2009). Based on Nalder and Wein (1999), we calculated an index of moss dominance (IMD) using the following Eq. (3):

$$IMD = \frac{O_{sph}}{O_{sph} + O_{pl} + O_{h} + O_{pt}}$$

where *O* is the sum of occurrence of a species in the 20 microplots (see section 2.1), *sph: Sphagnum* spp.; *pl: Pleurozium* **315** *schreberi* (Brid.) Mitt.; *h: Hylocomium splendens* (Hedw.) Schimp.; *pt: Ptilium crista-castrensis* (Hedw.).

Feather mosses dominate the moss stratum when the IMD tends toward 0, whereas *Sphagnum* spp. dominates the moss stratum when the IMD tends toward 1. Some sites (n = 5) that recently had fires did not have any moss species regrowth at the time of the fieldwork. For these sites, we set the IMD to 0.

B20 First, we wanted to estimate variation in the size of the bioreactive and recalcitrant soil C pools (Cfast and Cklow, respectively, expressed as stocks) with TSF. For each soil layer (O layer, top 15 cm of mineral soil and 15 to 35 cm of mineral soil), we scaled up to plot scale the cumulative proportion of C mineralized at the end of the incubations and the proportion of acid-

**2.4.2 Soil C quality and bioreactivity**

insoluble C using Eq. (4) and Eq. (5):

 $C_{fast} = \frac{C_{BioR}}{100} \times C \times D_B \times h$

 $325 \quad C_{slow} = \frac{C_{AI}}{100} \times C \times D_B \times h$

(3)

(4)

(5)

[revised manuscript text omitted]

|----|----------------------|
|    |                      |

|---|----------------------|

|----|----------------------|

| Supprime: I | H horizon |
|-------------|-----------|
|-------------|-----------|

effects on the CBioR of the mineral soil. TSF had a small positive direct effect on the CBioR of the mineral soil, but this

- 440 relationship was not significant (pc = 0.05, p > 0.36). In addition, pH induced two indirect effects on the CBioR of the mineral soil, i.e., through its negative and direct effects on Al and Mpy (pH $\rightarrow$ Al: pc = -0.24, p < 0.01; pH $\rightarrow$ Mpy: pc = -0.34, p < 0.01). Clay content had an indirect effect on the CBioR of the mineral soil, through its direct and positive effect on exchangeable Al (pc = 0.17 p < 0.05). Also, water balance had a weak indirect effect on the CBioR of the mineral soil through its direct effect on Mpy (pc = 0.11, p = 0.07).
- By allowing all the models (MIN1 and MIN2) to influence estimates, the model-averaging procedure indicated that the most important variables tested in this study and exerting a direct control over the CBioR of the mineral soil were as follows, by decreasing importance: exchangeable Al, metal oxide contents, pH and TSF (Table S2). Moreover, we failed to detect any direct effect of climate or mineral soil texture on CBioR.

**4 Discussion**

**450 4.1 Post-fire soil C quality**

Most of the studies on post-fire soil C have focused on immediate or short-term responses, and found that fire affects soil C quality by creating profound changes in the structure of soil organic matter compounds through thermal oxidation (Certini, 2005; Gonzalez-Perez et al., 2004). By using a long-term chronosequence of TSF ranging from two to 314 years, our study provides new insights into the understanding of the trajectory of changes in soil C quality following fire, over hundreds of

- 455 years. Our estimates of the size of fast- and slow-cycling soil C pools and our results indicate that *i*) both pools accumulate with TSF, and *ii*) the proportion of each C pool remains constant with TSF relative to total soil C stock (Fig. 2 and Table 1). These results do not necessarily imply that fire has no effect on soil C functional pools, because our chronosequence has a low resolution for the first few years following fire, but rather suggest that such changes, if present, are not long-lasting. Our results also highlight that the accumulation process of the bioreactive soil C reservoir does not reach an equilibrium, at least not in
- 460 the first three centuries following fire. Instead, environmental conditions limiting decomposition, such as cold temperatures under a thickening O layer developed with TSF, could have slowed down labile C degradation and allow its accumulation (Kane et al., 2005). Our results also emphasize that changes in the size of the soil functional reservoirs with TSF are stratified within the soil profile. This pattern is consistent with the fact that fire impact on soil C stock is limited to surface soil horizons (Andrieux et al., 2018).

**465 4.2 Control mechanisms of the soil carbon bioreactivity**

This study shows that soil  $C_{BioR}$  is driven by several climatic and non-climatic variables, some being common both for O layer and mineral soil, and others not, suggesting that different mechanisms may be involved in the control of the decomposition process in the O layer and in the mineral soil (Shaw et al., 2015; Ziegler et al., 2017).

**4.2.1 Soil carbon bioreactivity in the O layer**

- Our results suggested that pH and exchangeable Mn are important drivers of CBioR in the O laver. Boreal evergreen coniferous species generate high-lignin litter and forest floor layers (Laganière et al., 2017). This is reflected in the high proportion of acid-insoluble C of O layer samples (among all the O layer samples, mean ± sd = 73 ± 5%, Fig. S3). Therefore, soil C cycling in boreal forests depends on the capacity of microbes to depolymerize lignin. Microorganisms in the acidic soils of this ecosystem are dominated by fungi that use metalloenzymes–such as Mn peroxidases–to metabolize lignin (Pollegioni et al.,
- 2015), or are white-rot fungi (Basidiomycota) equipped with enzymes that oxidize lignin (Cragg et al., 2015). Soil C stocks in the boreal forest humus layer have been found to be negatively correlated with exchangeable Mn availability (Stendahl et al., 2017). In our study, exchangeable Mn of the O layer was positively correlated with CBioR, suggesting that increasing Mn

|---|------------------------------|

|----|--------------------------|
|    |                          |
|    |                          |
|    |                          |
| _  |                          |

availability stimulates organic matter breakdown and that an Mn bottleneck in soil C cycling may be present (Kranabetter, 2019). We also observed direct and positive causal relationships between pH and  $C_{BioR}$  of the O layer, indicating that acidic soil conditions limit soil C mineralization (Prescott et al., 2000). Bacterial respiration and microbial community composition

- 490 were found to be strongly determined by soil pH in the forest soil (Bååth and Anderson, 2003). We found that pH of the O layer decreased with TSF. Alongside the direct and positive effect of TSF on CBioR of the O layer, our results indicate that dynamic processes constrained by chemical soil properties shifting with stand development after burning (e.g. pH) drive the nature of soil organic matter and potentially the rate of C losses by heterotrophic respiration from boreal forest soils.
- Altogether, these results emphasize the need to include both soil chemistry and biological mechanisms into models of soil C
  cycling to better anticipate the role played by boreal forest in C cycle-climate feedbacks. Soil C cycling in mechanistic models of forest C dynamics often assumes that climate drives decay and the transfer rate of and between soil C pools (see Deluca and Boisvenue (2012)). Based on our results, we argue that chemical drivers of soil organic matter decomposition, such as exchangeable Mn concentrations and pH, might also be used to modulate soil C dynamics in such models, and we especially advise to accounting for temporal shifts in soil pH occurring with stand development.
- 500 We did not detect any direct effect of climate on soil CBioR in the O layer. This finding is consistent with the results of unchanged soil C stocks with *in situ* experimental warming worldwide (van Gestel et al., 2018). Furthermore, when synthesizing data of *in situ* experimental warming, Carey et al. (2016) found no change in soil respiration rate for warmed compared to control plots at the global scale, whereas changes were found to be significant for the boreal biome. The cumulative C mineralization of incubated soils in our study was not modulated by *in situ* emperature, which supports the results of Carey et al. (2016) for
- 505 their entire dataset, but not for the boreal biome-restricted dataset. However, Carey et al. (2016) did not study soils from the Canadian Boreal Shield.

**4.2.2 Soil carbon bioreactivity in the mineral soil**

As in the O layer, our results highlight the role of pH as a regulator of  $C_{BioR}$  in the mineral soil. In addition to having a direct effect on CBioR, pH also had two indirect effects. The first indirect effect is through the stimulation of metal oxide production 510 with increasing acidic conditions. We observed that low-pH conditions correlated positively with higher metal oxide contents, which in turn correlated negatively with CBioR in the mineral soil. This result is consistent with previous findings showing the role played by pH in mineral weathering and the preservation of C from decomposition through organo-metal complexation (Andrieux et al., 2018). The second indirect effect of pH on CBioR in the mineral soil is mediated through exchangeable Al only, not through Mn (Table S2). Microbes are vertically stratified within the soil column (Clemmensen et al., 2013; Ekschmitt 515 et al., 2008; Hynes and Germida, 2013), with fungi populating the upper soil layers because of their greater need for metabolic oxygen compared with bacteria, which can more easily dwell in the less-oxygenated deeper soil layers. Our results suggest that, contrary to the O layer, oxidative depolymerization of lignin compounds mediated by Mn peroxidases may not be a major process for C cycling in the mineral soil (see above). Instead, the negative effect of pH on exchangeable Al, together with the negative effect of exchangeable Al on CBioR in the mineral soil, indicates that low pH conditions favor a greater exchangeable 520 Al abundance, which in turn impedes organic matter decomposition. These findings are consistent with the observed pHdependent Al toxicity that slowed microbial catabolic activities in acidic forest soils in Japan (Kunito et al., 2016) and in laboratory experiments (Wood, 1995). Our study goes one step further in that we show that exchangeable Al content is directly

related to soil texture (especially clay content) in these podzolic soils. This supports the hypothesis that exchangeable Al bound to fine mineral particles, such as clay, might act as a source of stored Al that can be mobilized and complexed with C to impede
decomposition.

Contrary to the O layer, we found that TSF was only weakly correlated to mineral soil  $C_{BioR}$  and pH, and these relationships were not significant. We also found that the indirect effect of climate (correlation between water balance and metal oxides) on  $C_{BioR}$  was marginal. These results indicate that effects of TSF (direct and indirect) and water availability (indirect) on  $C_{BioR}$  are

|-----|-------------------------------------------------------------------------------------------------|
|     | Comparing to FITA                                                                               |
|     | Supprime: FH norizon                                                                            |
| - 1 |                                                                                                 |
|     |                                                                                                 |
|     |                                                                                                 |

[revised manuscript text omitted]
 balanc            | e (mm)                          | 493           | 660   | 558 (± 46)  |                      |                      |
| D (1 1                  | Sand                            |                      | 26    | 92          | 70 (± 11)     |                      |
| (%)                     | Silt                            |                      | 5     | 63          | 24 (± 10)     |                      |
|                         | Clay                            |                      | 2            | 22          | 6 (± 3)       |                      |
| Soil thickness          | s (cm)                   |                      | 10    | 103         | 39 (± 15)     |                      |
| Soil group a |                                 | 1                    |              | Podzol             |               | Tableau mis en forme |
|                         |                                 | All age-class (year) | 3.3   | 4.2         | 3.7 (± 0.2)   |                      |
|                         |                                 | [2;30]               | 3.4   | 4.1         | 3.9 (± 0.2)   |                      |
|                         |                                 | [30;60]       | 3.5   | 3.8         | 3.6 (± 0.1)   |                      |
|                         | O layer                  | [60;100[      | 3.3   | 4.2         | 3.7 (± 0.3)   |                      |
|                         |                                 | [100;150]            | 3.3   | 3.9         | 3.6 (± 0.2)   |                      |
|                         |                                 | [150;200]            | 3.5   | 3.9         | 3.7 (± 0.2)   |                      |
|                         |                                 | > 199      | 3.3   | 3.8         | 3.5 (± 0.2)   |                      |
|                         |                                 | All age-class (year) | 4.2   | 5.6         | 4.7 (± 0.2)   |                      |
|                         |                                 | [2;30]               | 4.4   | 5.0         | 4.7 (± 0.2)   |                      |
|                         |                                 | [30;60]              | 4.4          | 5.1         | 4.7 (± 0.2)   |                      |
| pH               | Mineral soil (top 15 cm)        | [60;100]             | 4.2          | 5.6         | 4.7 (± 0.3)   |                      |
|                         |                                 | [100;150]            | 4.4          | 4.9         | 4.6 (± 0.2)          |                      |
|                         |                                 | [150;200]            | 4.4   | 4.7         | 4.7 (± 0.3)   |                      |
|                         |                                 | > 199      | 4.3   | 5.2         | 4.6 (± 0.2)   |                      |
|                         |                                 | All age-class (year) | 4.5   | 5.9         | 5.2 (± 0.3)   |                      |
|                         |                                 | [2;30]               | 4.8   | 5.5         | 5.2 (± 0.2)   |                      |
|                         | Mineral soil (from 15 to 35 cm) | [30;60]              | 4.7          | 5.9         | 5.1 (± 0.4)   |                      |
|                         |                                 | [60;100]             | 4.6   | 5.8         | 5.3 (± 0.3)   |                      |
|                         |                                 | [100;150]            | 4.5          | 5.7         | 5.2 (± 0.3)   |                      |
|                         |                                 | [150;200]            | 4.9   | 5.5         | 5.2 (± 0.2)   |                      |
|                         |                                 | > 199                | 4.8   | 5.6         | 5.1 (± 0.2)   |                      |
|                         |                                 | All age-class (year) | 0.05         | 0.15               | 0.08 (± 0.02) |                      |
|                         |                                 | [2;30]               | 0.08         | 0.15               | 0.10 (± 0.02) |                      |
|                         |                                 | [30;60]              | 0.06  | 0.10               | 0.08 (± 0.01) |                      |
|                         | O layer                  | [60;100]             | 0.06  | 0.12               | 0.08 (± 0.02) |                      |
|                         |                                 | [100;150]            | 0.05         | 0.10        | 0.08 (± 0.01) |                      |
| Bulk                    |                                 | [150;200]            | 0.05         | 0.10               | 0.08 (± 0.02) |                      |
| density                 |                                 | > 199                | 0.06         | 0.10               | 0.08 (± 0.01)        |                      |
| (g.cm -3 )   |                                 | All age-class (year) | 0.73         | 1.60               | 1.06 (± 0.18)        |                      |
|                         |                                 | [2;30]               | 0.85         | 1.44               | 1.14 (± 0.15)        |                      |
|                         |                                 | [30;60]              | 0.83         | 1.32               | 1.11 (± 0.20)        |                      |
|                         | Mineral soil (top 15 cm)        | [60;100]             | 0.73         | 1. 60       | 1.02 (± 0.21)        |                      |
|                         |                                 | [100;150]            | 0.78         | 1.21               | 0.99 (± 0.14)        |                      |
|                         |                                 | [150;200]            | 0.83         | 1.30               | $1.07 (\pm 0.14)$    |                      |
| L                       | I                               | _ <del> </del>       |              |                    |                      |                      |

|                    |                                 | > 199      | 0.79        | 1.35 | 1.06 (± 0.16) |
|--------------------|---------------------------------|----------------------|-------------|-------------|----------------------|
|                    |                                 | All age-class (year) | 0.59 | 1.45 | 1.02 (± 0.21) |
|                    |                                 | [2;30]               | 0.67 | 1.29 | 1.03 (± 0.18) |
|                    |                                 | [30;60]              | 0.83 | 1.45 | 1.10 (± 0.20) |
|                    | Mineral soil (from 15 to 35 cm) | [60;100]             | 0.71 | 1.41 | 1.00 (± 0.23) |
|                    |                                 | [100;150]            | 0.59 | 1.33 | 0.98 (± 0.24) |
|                    |                                 | [150;200]            | 0.65 | 1.41 | 1.05 (± 0.21) |
|                    |                                 | > 199      | 0.62        | 1.32 | 0.98 (± 0.17) |
|                    |                                 | All age-class (year) | 9.8  | 49.3 | 22.6 (± 8.4)  |
|                    |                                 | [2;30]               | 9.8  | 36.1 | 17.2 (± 7.3)  |
|                    |                                 | [30;60]              | 11.6 | 20.9 | 17.2 (± 3.2)  |
| O layer depth (cm) |                                 | [60;100]             | 13.7 | 49.3 | 24.8 (± 9.7)  |
|                    |                                 | [100;150]            | 10.6 | 39.7 | 21.5 (± 8.3)  |
|                    |                                 | [150;200]            | 18.3 | 33.7 | 26.3 (± 4.8)  |
| Ţ                  |                                 | > 199      | 18.4        | 41.9        | 28.4 (± 6.8)  |

aAccording to IUSS Working Group WRB (2015).

Table 2: Post-fire soil C pool size and accumulation rates.

| Pool         | Unit                 | Layer                   | Mean (± sd)             | Equation                   | R²       | p-value           |
|--------------|-----------------------------|-------------------------|-------------------------|----------------------------|-----------------|-------------------|
|              | MgC.ha -1        | All                     | 150.80 (± 49.91) | $120.32 + 0.280 * TSF^{a}$ | 0.20     | < 0.001 |
| C tot |                             | O layer          | 80.37 (± 35.55)         | 66.00 + 0.13*TSF    | 0.09     | 0.011      |
|              |                             | Mineral soil (0-15 cm)  | 32.64 (± 14.72)  | 26.82 + 0.05 * TSF         | 0.08     | 0.013      |
|              |                             | Mineral soil (15-35 cm) | 39.05 (± 26.37)  | 29.22 + 0.09*TSF    | 0.08     | 0.020      |
|              |                             | All                     | 69.01 (± 25.79)  | 56.16 + 0.118*TSF   | 0.14     | 0.002      |
|              | McC hol                     | O layer          | 58.43 (± 26.82)  | 47.65 + 0.099*TSF          | 0.09     | 0.012      |
| C.           | MgC.na ·                    | Mineral soil (0-15 cm)  | 7.42 (± 3.95)    | 6.73 + 0.006*TSF    | 0.02            | 0.280             |
|              |                             | Mineral soil (15-35 cm) | 4.17 (± 3.61)    | 3.13 + 0.009 * TSF         | 0.04     | 0.076      |
| Cslow        | % of Ctot | All                     | 46.40 (± 9.35)   | 46.74 - 0.003*TSF   | < 0.01          | 0.808      |
|              |                             | O layer          | 72.58 (± 5.21)   | 71.97 + 0.006*TSF   | <0.01 | 0.469      |
|              |                             | Mineral soil (0-15 cm)  | 23.29 (± 7.80)   | 24.69 - 0.013*TSF   | 0.02     | 0.268      |
|              |                             | Mineral soil (15-35 cm) | 12.45 (± 9.19)   | 12.86 - 0.004*TSF   | <0.01 | 0.783      |
|              |                             | All                     | 11.47 (± 3.59)   | 9.25 + 0.020*TSF    | 0.21     | < 0.001 |
|              | MOLA                        | O layer          | 8.26 (± 3.86)    | 6.45 + 0.017*TSF    | 0.12     | 0.003      |
|              | Mgc.na ·                    | Mineral soil (0-15 cm)  | 2.06 (± 0.62)    | 1.86 + 0.002*TSF           | 0.05            | 0.050      |
| C            |                             | Mineral soil (15-35 cm) | 1.35 (± 0.79)    | 1.19 + 0.001 * TSF         | 0.02            | 0.214             |
| Lfast        |                             | All                     | 7.85 (± 1.67)    | 7.87 – 0.0002*TSF   | <0.01 | 0.935             |
|              | 0/ of C                     | O layer          | 10.55 (± 2.42)   | 10.11 + 0.004 * TSF        | 0.02            | 0.258             |
|              | 70 01 Ctot           | Mineral soil (0-15 cm)  | 7.16 (± 2.83)    | 7.42 - 0.002*TSF    | <0.01           | 0.590             |
|              |                             | Mineral soil (15-35 cm) | 4.19 (± 1.84)    | 4.33 - 0.001*TSF           | <0.01 | 0.646      |

| - · · · -                    |  |
|------------------------------|--|
| Supprime: ¶                  |  |

....

aTSF: time since fire (yr-1).

875

Table 3: Model fitness to the data for *a priori* hypotheses for the O layer. Models are sorted by increasing second-order Akaike information criterion (AICc)

W

 $Hypothesis \ Climate \ Texture \ C \ statistic \ df \ p \ K \ AIC_c \ \Delta AIC_c$

[revised manuscript text omitted]